# Diffusion Tree Sampling: Scalable inference-time alignment of diffusion models

**Vineet Jain**     **Kusha Sareen**     **Mohammad Pedramfar**     **Siamak Ravanbakhsh**
Mila – Québec AI Institute, McGill University
{jain.vineet,kusha.sareen,mohammad.pedramfar,siamak.ravanbakhsh}@mila.quebec

## Abstract

Adapting a pretrained diffusion model to new objectives at inference time remains an open problem in generative modeling. Existing steering methods suffer from inaccurate value estimation, especially at high noise levels, which biases guidance. Moreover, information from past runs is not reused to improve sample quality, resulting in inefficient use of compute. Inspired by the success of Monte Carlo Tree Search, we address these limitations by casting inference-time alignment as a search problem that reuses past computations. We introduce a tree-based approach that *samples* from the reward-aligned target density by propagating terminal rewards back through the diffusion chain and iteratively refining value estimates with each additional generation. Our proposed method, Diffusion Tree Sampling (DTS), produces asymptotically exact samples from the target distribution in the limit of infinite rollouts, and its greedy variant, Diffusion Tree Search (DTS$^\star$), performs a global search for high reward samples. On MNIST and CIFAR-10 class-conditional generation, DTS matches the FID of the best-performing baseline with up to $10\times$ less compute. In text-to-image generation and language completion tasks, DTS$^\star$ effectively searches for high reward samples that match best-of-N with up to $5\times$ less compute. By reusing information from previous generations, we get an *anytime algorithm* that turns additional compute into steadily better samples, providing a scalable approach for inference-time alignment of diffusion models. Project page: https://diffusion-tree-sampling.github.io.

## 1 Introduction

Diffusion models have emerged as one of the most powerful frameworks for generative modeling, achieving state-of-the-art results across a wide range of modalities, including image synthesis [30; 75; 64], molecule conformer generation [32; 89], and text generation [68; 48]. Despite their success, adapting a pretrained diffusion model to satisfy new, user-defined objectives at inference time without expensive retraining or fine-tuning remains a major challenge [79].

Most objectives can be cast as a reward function, turning alignment into a posterior sampling problem where the target is to sample from the pretrained model density weighted by exponentiated reward. The key challenge is that rewards are only available at the end of the denoising trajectory. So, inference-time alignment seeks to *guide the denoising process based on unseen terminal rewards*.

A range of different methods have been proposed – gradient-based guidance [17; 11; 3], where one uses reward gradients to bias the denoising process; sequential Monte Carlo (SMC) [84; 77; 7; 19; 39] which maintains a population of particles and resamples them during denoising based on an estimate of terminal rewards; or more recently, search-based methods [45; 46; 50] that perform a local greedy search based on approximate rewards. The common issue undermining all of these methods is that they rely on certain approximations to estimate the unseen terminal rewards. As we demonstrate in Section 3, these approximations bias decisions and degrade sample quality.

39th Conference on Neural Information Processing Systems (NeurIPS 2025).

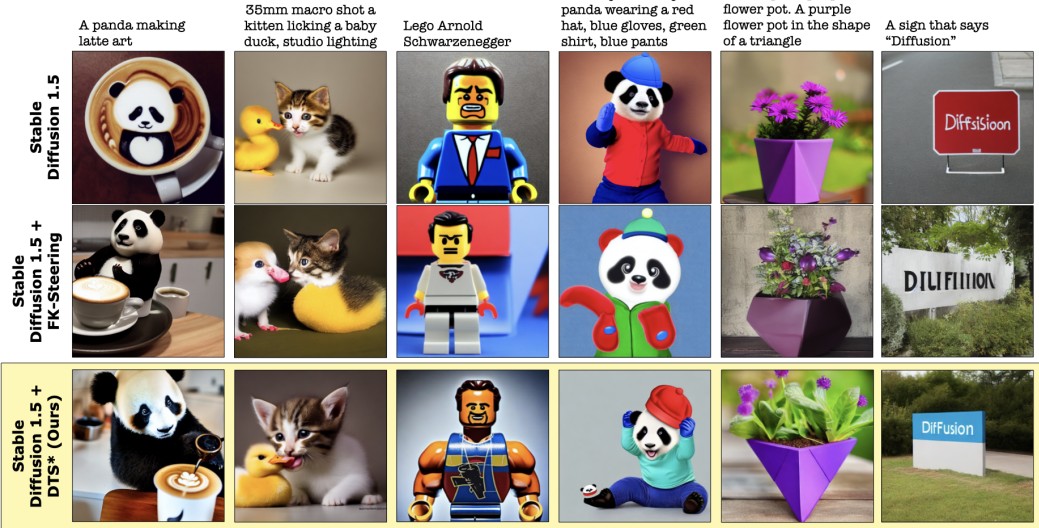

Figure 1: Sample text-image pairs using Stable Diffusion v1.5 [64] and ImageReward [88] as the guiding function, with generated samples picked at random for each method and prompt.

We therefore address the following challenges or questions in this work: (1) how to guide the diffusion process at inference-time when rewards are available only at the end? This is also known as the credit assignment problem in reinforcement learning (RL) literature [51]; (2) inference-time samples can potentially inform and improve future samples – how to systematically use this information in a sequential yet scalable sampling process?

Fortunately, RL also provides a solution that has been historically quite successful in addressing both challenges – Monte Carlo Tree Search (MCTS) [6]. We therefore ask: can we leverage MCTS for steering diffusion models? We observe that during denoising, the pretrained diffusion model can be viewed as a deterministic policy, while the reverse process Gaussian step can be viewed as a stochastic environment transition. This is exactly the classical setting for MCTS [41], suggesting we could use tree search to solve the problem of inference-time alignment.

Our proposed algorithm, Diffusion Tree Sampling (DTS) is a novel inference-time alignment method that casts the denoising process as a finite-horizon tree, where similar to MCTS, rollouts are used to continuously improve value estimates for intermediate noisy states. For applications that require optimization, rather than sampling from the target density, we propose a search variant – Diffusion Tree Search (DTS$^\star$) – that performs a principled search in the space of denoising trajectories to identify the modes within high-volume regions of the target density.

Our contributions can be summarized as follows:

- We formulate inference-time alignment of diffusion models as a tree search problem for sampling from the reward-aligned distribution or optimizing for high reward samples.
- We develop a general tree-based algorithm that yields asymptotically exact samples from the target distribution in the limit of infinite rollouts.
- We demonstrate that DTS significantly reduces bias and variance in value estimation compared to common approximations used by many existing methods.
- We show that both DTS and DTS$^\star$ scale more favorably compared to leading baselines and match their performance with up to $10\times$ less compute on class-conditional image generation, and up to $5\times$ less compute on text-to-image alignment and language completion tasks.

## 2 Preliminaries

**Diffusion models.**   Diffusion models [30; 75] define a generative process via a Markov chain that progressively adds noise to data $\mathbf{x}_0 \sim p_{\text{data}}(\mathbf{x})$, referred to as the forward process, $\mathbf{x}_t = \sqrt{\alpha_t}\,\mathbf{x}_{t-1} + \sqrt{1-\alpha_t}\,\epsilon, \quad \epsilon \sim \mathcal{N}(0, I)$, where $t \in \{1, \ldots, T\}$ indexes discrete time steps, and $\{\alpha_t\}_{t=1}^{T}$ defines a noise schedule. The noise schedule is chosen such that at $t = T$, the marginal distribution of the samples resembles a simple fixed distribution, such as standard Gaussian, $p(\mathbf{x}_T) = \mathcal{N}(0, I)$.

A learned reverse process iteratively denoises samples: $p_\theta(\mathbf{x}_{t-1} \mid \mathbf{x}_t) = \mathcal{N}(\mathbf{x}_{t-1}; \mu_\theta(\mathbf{x}_t, t), \sigma_t^2 I)$, where $\sigma_t$ is the posterior variance calculated from the forward noise schedule and $\mu_\theta$ is parameterized typically by neural networks and optimized by minimizing the variational bound on the data likelihood or via denoising score matching. The generative process induces a distribution:

$$p_\theta(\mathbf{x}_0, \ldots, \mathbf{x}_{T-1}, \mathbf{x}_T) = p(\mathbf{x}_T) \prod_{t=1}^{T} p_\theta(\mathbf{x}_{t-1} \mid \mathbf{x}_t), \quad p(\mathbf{x}_T) = \mathcal{N}(0, I). \tag{1}$$

**Alignment of diffusion models.** Consider a pretrained diffusion model $p_\theta$ and an optimality variable $\mathcal{O} \in \{0, 1\}$ which denotes whether a sample $\mathbf{x} \sim p_\theta(\mathbf{x})$ satisfies some desirable property. This is equivalent to sampling from the posterior distribution $p(\mathbf{x} \mid \mathcal{O} = 1) \propto p_\theta(\mathbf{x}) \, p(\mathcal{O} = 1 \mid \mathbf{x})$. A typical assumption is that $p(\mathcal{O} = 1 \mid \mathbf{x}) \propto \exp(\lambda r(\mathbf{x}))$ where $r$ is some reward function and $\lambda$ is the inverse temperature. For the rest of this paper, we assume that $\lambda = 1$ unless otherwise stated and we define the alignment problem as sampling from the target distribution or finding its mode, where $Z$ is the normalization constant:

$$\pi^*(\mathbf{x}) = \frac{1}{Z} \, p_\theta(\mathbf{x}) \exp(\lambda r(\mathbf{x})). \tag{2}$$

**Reinforcement learning approach.** Since the generative process in diffusion models defines a Markov chain, we may consider the model $p_\theta$ as a policy. The target distribution $\pi^*$ can be seen as the optimal policy for the following objective:

$$\pi^*(\mathbf{x}) := \underset{\pi}{\mathrm{argmax}} \, \mathbb{E}_{\mathbf{x} \sim \pi(\cdot)} \left[ r(\mathbf{x}) \right] - \frac{1}{\lambda} \, D_{\mathrm{KL}} \left( \pi \, \| \, p_\theta \right). \tag{3}$$

This is closely related to the maximum entropy RL objective [93; 22], except that the entropy regularization is replaced by the KL divergence with the pretrained model $p_\theta$. We define the *soft value function* at timestep $t$ as the expected exponentiated reward starting from $\mathbf{x}_t$ and following $p_\theta$:

$$V_t(\mathbf{x}_t) := \frac{1}{\lambda} \log \mathbb{E}_{p_\theta(\mathbf{x}_{0:t-1} \mid \mathbf{x}_t)} \left[ \exp \left( \lambda r(\mathbf{x}_0) \right) \right]. \tag{4}$$

This soft value function satisfies the following recursive relation, analogous to the soft Bellman equation and exactly characterizes the optimal policy $\pi^*$:

$$V_t(\mathbf{x}_t) = \frac{1}{\lambda} \log \mathbb{E}_{p_\theta(\mathbf{x}_{t-1} \mid \mathbf{x}_t)} \left[ \exp \left( \lambda V_{t-1}(\mathbf{x}_{t-1}) \right) \right], \quad V_0(\mathbf{x}_0) = r(\mathbf{x}_0). \tag{5}$$

$$\pi_t^*(\mathbf{x}_{t-1} \mid \mathbf{x}_t) = \frac{p_\theta(\mathbf{x}_{t-1} \mid \mathbf{x}_t) \exp \left( \lambda V_{t-1}(\mathbf{x}_{t-1}) \right)}{\int p_\theta(\mathbf{x}_{t-1} \mid \mathbf{x}_t) \exp \left( \lambda V_{t-1}(\mathbf{x}_{t-1}) \right) d\mathbf{x}_{t-1}}. \tag{6}$$

This formulation explicitly connects optimal sampling with soft value estimation, motivating various practical approximations and sampling methods discussed in subsequent sections. For completeness, we derive Equations (5) and (6) in Section D.1. In the rest of the paper, we use $V_t$ to denote the true soft value function and $\hat{v}_t$ to denote estimates.

## 3 Inference-time adaptation of diffusion models

One option to obtain the optimal policy in Equation (6) is to fine-tune the diffusion model using RL [21; 5; 81; 18]. Sampling from unseen reward functions would require guiding the denoising process during inference to align with the optimal policy without modifying the prior pretrained model. We discuss some of the most relevant works below, and works in related areas in Section A.

**Gradient-based guidance.** One way to sample from the optimal policy $\pi^*$ is to use the first-order Taylor expansion of $V_{t-1}$ around the pretrained mean $\mu_\theta(\mathbf{x}_t, t)$. This yields the gradient-based denoising step $\tilde{\mathbf{x}}_{t-1} \sim \mathcal{N} \left( \mu_\theta(\mathbf{x}_t, t) + \lambda \sigma_t^2 \nabla_{\mathbf{x}_{t-1}} V_{t-1}(\mathbf{x}_{t-1}), \sigma_t^2 I \right)$. This can be considered a form of classifier guidance [17] and is used in many proposed inference-time steering methods [11; 3; 27]. The gradient approximation can be improved by using Monte Carlo samples for estimation [74].

**Sequential Monte Carlo.** Particle-based methods are another very popular approach, where a population of samples is maintained to approximately sample from the desired distribution. Sequential Monte Carlo (SMC) [15] uses potential functions, which usually approximate the soft value function, to assign weights to particles and resample them at every step. Different variations of SMC have been proposed for diffusion model alignment [84; 77; 7; 19; 39; 72]. Classical SMC guarantees exact sampling in the limit of infinite particles and exact value estimation. In practice, however, the repeated sampling procedure can reduce diversity due to weight variance and inaccurate value estimates. We provide detailed background on SMC for diffusion sampling in Section B.

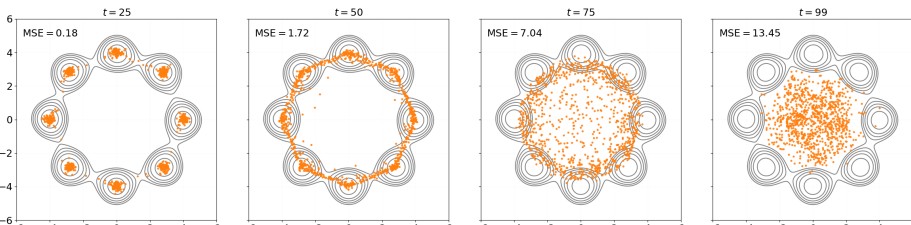

Figure 2: One-step prediction $\hat{\mathbf{x}}_0(\mathbf{x}_t)$ using Tweedie's formula for different time steps, along with average mean squared error with the ground truth data samples. Close to $t = 0$, the predictions are fairly accurate, but towards the maximum timestep $T = 99$, they devolve into random predictions.

**Search-based methods.** Recently, there has been a growing interest in using search-based methods to align diffusion models [50]. Most of these methods propose doing a local search [45; 46] by obtaining multiple denoising candidates at each step and selecting the best one based on their value. More recently, tree search has been combined with best-of-N [92], and an MCTS-based approach [91] has been applied in the specific context of diffusion forcing [9] over sequences for planning. However, these methods either do not use an explicit backup mechanism, resulting in a limited local search[45; 46; 92; 50], or they rely on inaccurate value estimates [91]. DTS, on the other hand, performs global credit assignment using all trajectories for asymptotically exact sampling.

### 3.1 The value estimation problem

Most existing methods use soft values from Equation (5) either by taking their gradient [11; 74], as SMC potentials [84; 39], or for search [45; 46; 91]. Since estimating these soft values is intractable, they employ the following set of approximations. The first step is to apply Jensen's inequality:

$$V_t(\mathbf{x}_t) = \frac{1}{\lambda} \log \mathbb{E}_{p_\theta(\mathbf{x}_{0:t-1}|\mathbf{x}_t)} \left[ \exp\left( \lambda\, r(\mathbf{x}_0) \right) \right] \approx \mathbb{E}_{p_\theta(\mathbf{x}_{0:t-1}|\mathbf{x}_t)} \left[ r(\mathbf{x}_0) \right] \tag{7}$$

Estimation of this expectation is computationally expensive since it requires multiple rollouts from the current sample at timestep $t$ to the clean sample. The expected reward is further approximated as $\mathbb{E}_{p_\theta(\mathbf{x}_{0:t-1}|\mathbf{x}_t)} \left[ r(\mathbf{x}_0) \right] \approx r(\hat{\mathbf{x}}_0(\mathbf{x}_t))$, where $\hat{\mathbf{x}}_0$ is the posterior mean obtained using Tweedie's formula [20; 11] in a single step:

$$\hat{\mathbf{x}}_0(\mathbf{x}_t) = \mathbb{E}_{p_\theta(\mathbf{x}_{0:t-1}|\mathbf{x}_t)} \left[ \mathbf{x}_0 \right] = \frac{1}{\sqrt{\alpha_t}} \left( \mathbf{x}_t + (1 - \bar{\alpha}_t) \nabla_{\mathbf{x}_t} \log p_t(\mathbf{x}_t) \right). \tag{8}$$

The posterior mean is an approximation because the true score function for intermediate marginal densities $\nabla_{\mathbf{x}_t} \log p_t(\mathbf{x}_t)$ is replaced by the learned score function. We investigate the effect of this approximation for a diffusion model trained on a mixture of Gaussians in Figure 2, where we see that the prediction based on Tweedie's formula gets increasingly inaccurate for higher noise levels. Therefore, despite the wide adoption of this approximation, the value estimates used for guidance are essentially random at higher noise levels even in simple 2D settings.

### 3.2 Scaling with compute

Efficient utilization of available compute is critical for any inference-time alignment algorithm. Existing SMC or search-based methods treat each sampling procedure as an independent event, and all intermediate evaluations are discarded. Consider a streaming or repeated sampling setting. There is no mechanism to assimilate information from prior runs to improve sample quality. This could be particularly useful for correcting errors in value estimation, which, as we saw above, is difficult at high noise levels. In other words, these methods scale *parallelly* by increasing particle count, but do not scale *sequentially* by turning extra compute into cumulative improvements in estimate quality.

## 4 Diffusion Tree Sampling and Search

The pitfalls above suggest two complementary desiderata for an effective inference–time sampler:

- (D1) Use information from low-noise timesteps, where the reward signal is reliable, to *refine decisions made at high-noise timesteps*, rather than treating every step in isolation.

- (D2) Reuse information from previously explored trajectories so that *additional compute improves sample quality* instead of merely increasing parallel particle count (this property is characteristic of an anytime algorithm).

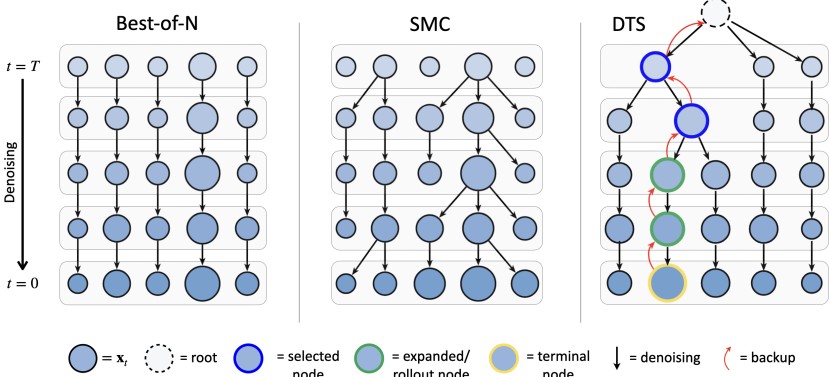

Figure 3: Illustration of various inference-time steering methods, where size of the node represents the associated values. **Left:** Best-of-N denoises multiple samples using the base diffusion model and selects the one with the highest reward. **Center:** SMC maintains a population of particles and resamples based on an estimate of the value function. **Right:** DTS and DTS$^\star$ maintain a tree that accumulates information across multiple rollouts and backs up the terminal reward to refine value estimates. The diagram illustrates the four phases: selection, expansion, rollout, and backup.

To address these issues, in this section we develop a solution by first interpreting the denoising process as a tree in Section 4.1. We then introduce a general tree-based algorithm to sample from the target density in Section 4.2 and describe its application to diffusion model alignment in Section 4.3. Finally, in Section 4.4 we empirically evaluate our method on 2D datasets to validate, in a very clear and controlled setting, whether DTS satisfies the desiderata mentioned above.

## 4.1 Denoising tree

The Markov property of the reverse diffusion chain naturally induces a finite horizon tree in $\mathbb{R}^d$, where $d$ is the dimensionality of the space over which we are diffusing. Here, the nodes at depth $t$ represent noisy states $\mathbf{x}_t$ and the edges represent a denoising step. Each node $\mathbf{x}_t$ can be stochastically denoised into multiple children $\mathbf{x}_{t-1} \sim p_\theta(\cdot \mid \mathbf{x}_t)$.

This framing allows us to keep track of information across multiple denoising trajectories, including estimates of the soft value function, which helps with global credit assignment. Using the tree structure gives us the flexibility to sample from the target density or search for the highest reward sample with minimal changes to the underlying algorithm. We call the sampling variant *Diffusion Tree Sampling* (DTS) and the search variant *Diffusion Tree Search* (DTS$^\star$).

## 4.2 Tree-based sampling

Similar to MCTS, we construct a tree $\mathcal{T}$, where nodes represent states $\mathbf{x}_t$ and edges represent transitions $p_\theta(\mathbf{x}_{t-1} \mid \mathbf{x}_t)$ following the base diffusion model. Each node maintains the current state and timestep $(\mathbf{x}_t, t)$, an estimate of the soft value function $\hat{v}(\mathbf{x}_t)$, and the visit count $N(\mathbf{x}_t)$. Since we do not have a fixed starting state, we introduce a dummy state as the root $\mathbf{x}_{T+1}$ that transitions to the prior in our diffusion model – i.e., $p(\mathbf{x}_T \mid \mathbf{x}_{T+1}) = \mathcal{N}(\mathbf{0}, I)$. Additionally, we use $\mathcal{C}(\mathbf{x}_t)$ to denote the set of children of node $\mathbf{x}_t$.

The goal is to expand this tree while improving value estimates as we expand it, so that it can be used for approximate sampling from the target distribution at any time during the construction. The resulting tree sampling process provably samples from the target distribution $\pi^*$ in the limit of infinite rollouts. The tree-building procedure of DTS repeats the following steps iteratively:

1. **Selection.** Starting from the root $\mathbf{x}_{T+1}$, sample a child $\mathbf{x}_{t-1} \in \mathcal{C}(\mathbf{x}_t)$ from Boltzmann distribution $\propto \exp\left(\lambda\, \hat{v}(\mathbf{x}_{t-1})\right)$ recursively until either an unexpanded node is reached or $t = 0$.

2. **Expansion.** If we reach a node $\mathbf{x}_t$ such that the number of children is less than the maximum allowed value and $t > 0$, we create a new child node $\mathbf{x}_{t-1} \sim p_\theta(\cdot \mid \mathbf{x}_t)$ and initialize $\hat{v}(\mathbf{x}_{t-1}) = 0, \ N(\mathbf{x}_{t-1}) = 1$.

3. **Rollout.** From the newly created node, we perform a rollout till terminal states $\mathbf{x}_0$ by recursively sampling from $p_\theta(\cdot \mid \mathbf{x}_{t'})$ for $t' = t - 1, \ldots, 0$. An important distinction from traditional MCTS is that we add the rollout path to $\mathcal{T}$.

4. **Backup.** Evaluate the terminal node using the reward function $\hat{v}(\mathbf{x}_0) = r(\mathbf{x}_0)$ and use soft Bellman equation (Equation (5)) to update parent node values using the children node values recursively for $t = 0, \ldots, T$. The visit counts for all nodes in the path are also updated.

Each traversal of the tree, from the root to the backup of the value function, constitutes one tree-building iteration. For sampling from $\mathcal{T}$, we simply start from the root and perform selection steps until we reach a terminal node. A formal algorithm is provided in Section E.

**Proposition 1** (Asymptotic consistency). *Let $r$ be bounded and $\lambda > 0$, then DTS produces a sequence of terminal states whose empirical distribution converges to the optimal policy $\pi^*$ as the number of tree iterations $M \to \infty$.*

*Proof sketch.* By construction, the tree policy selects $\mathbf{x}_{t+1}$ with unnormalized probability $p_\theta(\mathbf{x}_{t+1} \mid \mathbf{x}_t) \exp(\lambda \hat{v}(\mathbf{x}_{t+1}))$, which is the optimal policy defined in Equation (6). By telescoping the product over $t$, we obtain the final samples at $t = T$ are sampled from $p_\theta(\mathbf{x}) \exp(r(\mathbf{x}_0))$. A more detailed proof is given in Section D. $\qquad\square$

### 4.3 Design choices for diffusion alignment

The algorithm discussed above can be applied to any Markov chain. However, in this work, we apply it to the problem of inference-time alignment of diffusion models. We discuss various considerations and design choices below, with more implementation details in Section F.

**Sampling or Search.** DTS is designed to sample from the target distribution $\pi^*$, but for settings where a single high-quality sample is required, we introduce a *search* variant, DTS$^\star$. It keeps the same soft value backup but modifies the selection step by always selecting the child with the largest soft value estimate instead of Boltzmann sampling. Since DTS$^\star$ uses soft values, this is different from standard MCTS – it implements a *marginal-MAP* (max–sum) inference scheme [63] over the tree. At every noise step, the algorithm selects the branch whose subtree carries the greatest mass under $\pi^\star$ and, once $t = 0$ is reached, returns the highest-density leaf inside that dominant region. As we will see in the Section 5, this volume-based selection helps avoid reward over-optimization.

**Branching.** Extensions of MCTS to continuous spaces commonly use *progressive widening* [14] to decide the maximum number of branches $B(\mathbf{x}_t)$ allowed per node based on the number of visits: $B(\mathbf{x}_t) = C \cdot N(\mathbf{x}_t)^\alpha$, $C > 0$, $\alpha \in (0, 1)$. The high-level intuition is that nodes that are visited more often should be expanded more, since they represent more promising directions for denoising. We adopt the same strategy and during tree traversal, if we encounter a node such that $|\mathcal{C}(\mathbf{x}_t)| < B(\mathbf{x}_t)$ and $t > 0$, we will always expand.

**Exploration.** There is a rich literature on search methods for classical MCTS, and the most popular approach, UCT [41], is an application of upper-confidence bounds [2] to trees. We employ this exploration strategy for DTS$^\star$, i.e. we choose the child $\mathbf{x}_{t-1} \in \mathcal{C}(\mathbf{x}_t)$ with the maximum value of the UCT estimate:

$$\mathrm{UCT}(\mathbf{x}_{t-1}) = \hat{v}(\mathbf{x}_{t-1}) + c_{\mathrm{uct}} \sqrt{\frac{\log N(\mathbf{x}_t)}{N(\mathbf{x}_{t-1})}}, \qquad c_{\mathrm{uct}} > 0. \tag{9}$$

For DTS, we do not employ explicit exploration, because, in practice we observe that sampling obviates the need for an exploration bonus or handcrafted mechanism. This is also supported theoretically by recent work in the bandit setting [57].

**Efficient implementation.** The main computational cost is incurred when using the diffusion model proposal to sample new children or perform rollouts. We implement an efficient batched version of the algorithm by collecting nodes in a batch and performing a single batched denoising rollout per iteration. The selection and backup steps involve simple tensor operations and pointer manipulation with negligible cost. Therefore, while the control flow of our method is sequential, the practical algorithm can be parallelized. Note that once the tree has been built, sampling is near instantaneous by repeatedly selecting children without any model calls.

### 4.4 Illustrative experiments

In this section, we perform experiments on simple 2D settings to answer the following questions:

- Does DTS sample accurately from the target distribution?
- Does reward backup in DTS result in more accurate value estimates (desideratum D1)?
- Does sample quality of DTS improve with more inference-time compute (desideratum D2)?

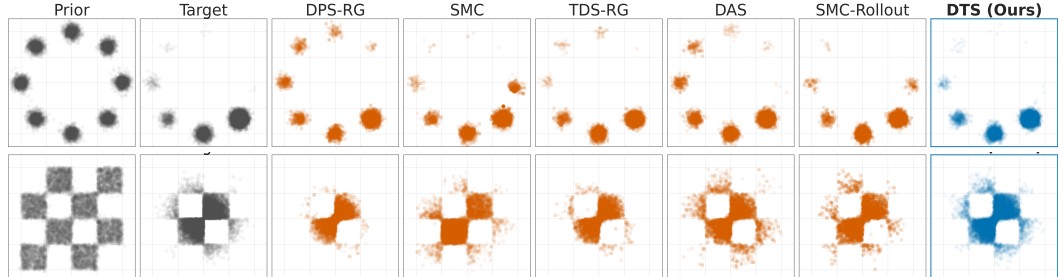

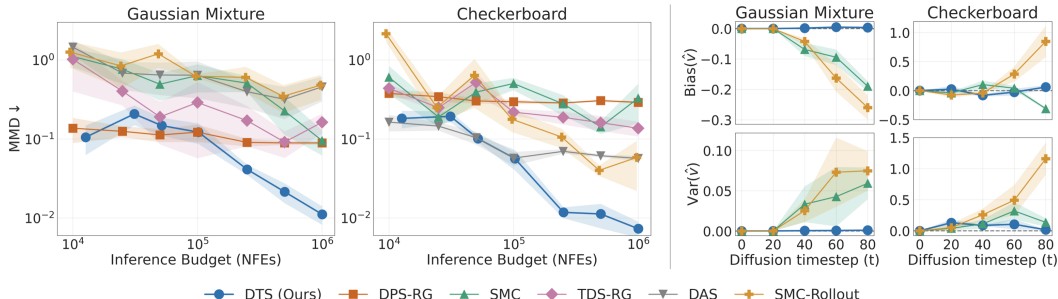

Figure 4: Samples from the prior $p(\mathbf{x}_0)$, target $p(\mathbf{x}_0)\exp(r(\mathbf{x}_0))\,/\,\mathcal{Z}$ and different sampling methods at $10^6$ NFEs. **Top:** The prior is an equal-weighted mixture of Gaussians, and the reward function distributes mass unevenly. **Bottom:** The prior has support on alternate square regions in a checkerboard pattern, and the reward function $r(x, y) = -0.5(x^2 + y^2)$ is negative distance from the origin. More details on the experimental setup are provided in Section F.2.

Figure 5: **Left:** Maximum mean discrepancy (MMD) between generated samples and target ground truth samples as a function of number of function evaluations of the prior diffusion model. **Right:** Bias and variance of value estimates for different approaches at $10^6$ NFEs.

**Setting.** We compare DTS with several inference-time steering methods including some which were originally proposed for posterior sampling in inverse problems, and adapt them to the reward-guidance setting: (1) *DPS-RG* (our reward-guided version of DPS [11]) = gradient-based guidance only, (2) *SMC* [71], (3) *TDS-RG* (reward-guided version of TDS [84]) = SMC + gradient guidance, (4) *DAS* [39] = SMC + gradient guidance + tempering. We also implement a version of SMC, which we call *SMC-Rollout*, where the values are estimated via one full DDIM [73] rollout. For fair comparison, we benchmark with respect to the number of function evaluations (NFEs) of the diffusion model.

**Results.** Figure 4 plots the samples obtained using different methods for two different settings. In both cases, DTS approximates the ground truth target density more accurately, with other methods distributing mass inaccurately to different areas of the support. In particular, gradient-based methods like DPS-RG and TDS-RG suffer from instability and require gradient clipping to stabilize denoising steps. Figure 5 shows the Maximum mean discrepancy (MMD) between generated samples and ground truth samples for an increasing amount of inference budget, measured in terms of the number of function evaluations of the diffusion model. The results empirically validate that the sample quality of DTS improves with more compute, satisfying desideratum D2.

**Bias-variance analysis of value estimates.** To verify if DTS leads to more accurate soft value estimates, we perform a bias-variance analysis. We estimate ground truth soft value estimates based on 1000 rollouts from noisy states and then compute the bias and variance of the value estimates. Figure 5 compares DTS, Tweedie's formula (SMC + variants), and a single full DDIM rollout (SMC-Rollout). Both one-step denoising and single rollout have high bias and variance, which generally get worse for higher timesteps. Our tree-based approach reduces bias by using accurate reward information and reduces variance by aggregating information from multiple rollouts. This empirically validates that DTS satisfies desideratum D1.

## 5 Experiments

We validate the efficacy of DTS and DTS$^\star$ for image and text generation. Our experiments show that DTS draws faithful samples from high-dimensional posteriors, and DTS$^\star$ efficiently searches high-dimensional image space to discover high-reward samples. We also test our method on image

Table 1: Comparison of inference-time posterior sampling methods. We report the mean±std of each metric across the relevant classes and highlight ±5% values from the best experimental value.

| Dataset → | MNIST | | | | MNIST even/odd | | | | CIFAR-10 | | | |
|---|---|---|---|---|---|---|---|---|---|---|---|---|
| Algorithm ↓ | FID (↓) | CMMD (↓) | $\mathbb{E}[\log r(\mathbf{x})]$ (↑) | Diversity (↑) | FID (↓) | CMMD (↓) | $\mathbb{E}[\log r(\mathbf{x})]$ (↑) | Diversity (↑) | FID (↓) | CMMD (↓) | $\mathbb{E}[\log r(\mathbf{x})]$ (↑) | Diversity (↑) |
| DPS | 0.359 ±0.227 | 0.441 ±0.447 | -0.323 ±0.286 | 0.474 ±0.051 | 0.123 ±0.031 | 0.293 ±0.139 | -0.002 ±0.001 | 0.572 ±0.053 | 0.486 ±0.121 | 2.609 ±0.824 | -0.002 ±0.001 | 0.551 ±0.024 |
| SMC/FK | 0.060 ±0.051 | 0.177 ±0.142 | -0.002 ±0.004 | 0.422 ±0.040 | 0.027 ±0.009 | 0.123 ±0.113 | -0.003 ±0.003 | 0.583 ±0.084 | 0.313 ±0.070 | 1.409 ±0.445 | -0.102 ±0.093 | 0.487 ±0.045 |
| TDS | 0.087 ±0.035 | 0.463 ±0.260 | -0.001 ±0.001 | 0.404 ±0.042 | 0.053 ±0.010 | 0.250 ±0.056 | -0.001 ±0.000 | 0.576 ±0.124 | 0.487 ±0.112 | 2.675 ±0.665 | -0.046 ±0.055 | 0.469 ±0.042 |
| DAS | 0.039 ±0.017 | 0.179 ±0.099 | -0.016 ±0.016 | 0.440 ±0.041 | 0.031 ±0.002 | 0.079 ±0.011 | -0.015 ±0.019 | 0.603 ±0.094 | 0.241 ±0.037 | 0.822 ±0.203 | -0.584 ±0.200 | 0.530 ±0.023 |
| **DTS (ours)** | 0.014 ±0.005 | 0.068 ±0.030 | -0.023 ±0.006 | 0.452 ±0.050 | 0.007 ±0.003 | 0.036 ±0.029 | -0.010 ±0.004 | 0.597 ±0.069 | 0.195 ±0.041 | 0.745 ±0.201 | -0.305 ±0.116 | 0.542 ±0.020 |

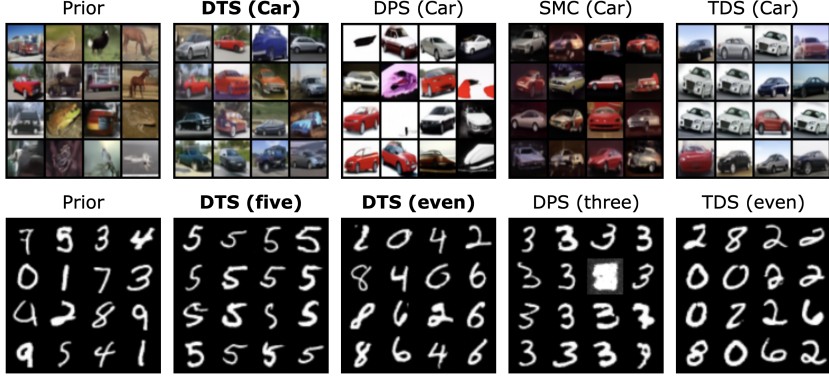

Figure 6: FID (lower is better) versus number of function evaluations for different methods on MNIST single digit generation averaged over all 10 digits (left), MNIST odd and even digit generation (center), and CIFAR-10 single class generation averaged over all 10 classes (right). All methods were evaluated with 5000 generated samples per class.

inpainting task in Section I, provide detailed memory/wall-clock time analysis in Section H, and some additional results in Section J. A detailed description of each experimental setting is provided in Section F.2, and details of baseline implementations are in Section G.

## 5.1 Class-conditional posterior sampling

We evaluate DTS on the task of sampling from a class-conditioned posterior distribution $p(\mathbf{x} \mid c) \propto p_\theta(\mathbf{x})p(c \mid \mathbf{x})$ where $p_\theta(\mathbf{x})$ is a pretrained unconditional diffusion model and $p(c \mid \mathbf{x})$ is a classifier. This would correspond to setting $r(\mathbf{x}) = \log p(c \mid \mathbf{x})$ in Equation (2).

**Setting.** We use MNIST and CIFAR-10 datasets, each with 10 classes. In both cases, the priors are unconditional diffusion models in pixel-space – we train one from scratch on MNIST and use an off-the-shelf model for CIFAR-10. For MNIST, we consider two settings: sampling from individual digits, and sampling from even/odd digits. The latter is a multimodal posterior with reward function $r(\mathbf{x}) = \max_{\{i=0,2,4,6,8\}} \log p(c = i \mid \mathbf{x})$ for the even digits and similarly for odd digits. For CIFAR-10, we sample from individual classes.

Figure 7: Samples generated from the CIFAR-10 (top) and MNIST (bottom) base diffusion models, and posterior samples using different methods at $10^6$ NFEs. Gradient-based guidance such as DPS can be unstable, leading to samples that lie outside the support of the prior. SMC-based methods struggle to accurately sample from multi-modal distributions – for MNIST even digits, TDS oversamples from the digit two and undersamples from the digit four, and for CIFAR-10 car, both SMC and TDS suffer from mode collapse.

**Baselines.** We compare the performance of DTS with DPS [11], SMC/FK [71], TDS [84] and DAS [39]. DPS, TDS, and DAS use reward gradients, while SMC/FK is derivative-free. We report two distribution-based metrics – Fréchet Inception Distance (FID) and CLIP maximum mean discrepancy (CMMD) [37] – that compare generated samples with ground truth samples from the dataset, in addition to average rewards and CLIP diversity (pairwise cosine distance).

**Results.** Table 1 reports the mean±std of various metrics for different methods after $10^6$ NFEs. In all three settings, DTS achieves the lowest FID and CMMD by a considerable margin, indicating it closely matches the true posterior. We observe that the margin of improvement on CIFAR-10 narrows slightly. We attribute this to reward noise: the CIFAR-10 classifier achieves an accuracy of $\sim 85\%$, so its logits provide a noisier signal than the near-perfect classifier used on MNIST. Even so, DTS still outperforms all baselines. TDS and SMC in particular show characteristics of mode collapse with very high average rewards and low diversity, whereas DPS often generates samples that lie outside the support of the base model. Figure 6 shows that DTS achieves very low FID across different NFEs and has better scaling properties with more compute compared to existing methods. Figure 7 presents example outputs from each method, highlighting their specific characteristics. We present samples for all classes and plots of additional metrics as a function of NFEs in Section J.2.

## 5.2 Text-to-image generation

**Setting.** We use Stable Diffusion v1.5 [64], a latent diffusion model, as the prior over $512 \times 512$ images $\mathbf{x} \sim p_\theta(\mathbf{x} \mid \mathbf{y})$ where $\mathbf{y}$ denotes the text prompt. We evaluate on two different settings: (a) DrawBench [67], which is comprehensive benchmark of 200 prompts, with ImageReward [88] $r(\mathbf{x}, \mathbf{y})$ that encodes prompt accuracy as well as human preferences; and (b) following Black et al. [5], we use 45 common animals from the ImageNet-1000 dataset as prompts, with the LAION aesthetics predictor [69] $r(\mathbf{x})$ that encodes aesthetic quality of an image but does not check for prompt accuracy.

**Baselines.** A strong baseline for high-reward generation is *best-of-N*, which draws $N$ samples from the base model and keeps the one with the highest reward. SMC has also been applied to this problem [71], but (a) as discussed in Section 3.1, it relies on one-step value estimates that become inaccurate at high noise, and (b) Section 5.1 shows that it often collapses onto narrow modes. We compare $\text{DTS}^\star$ with best-of-N and FK-Steering (SMC) [71]. We compare all methods in a compute-matched setting, measured in terms of the number of diffusion model calls (NFEs).

**Results.** Figure 8 plots the maximum ImageReward and aesthetic score versus inference compute. $\text{DTS}^\star$ outperforms the baselines for DrawBench prompts, see Figure 1 for examples. One strong feature of $\text{DTS}^\star$ is its favorable scaling properties with more compute. In the aesthetic score setting, FK/SMC achieves the highest rewards across NFEs, but we see severe over-optimization on all prompts. $\text{DTS}^\star$ manages to strike the right balance between achieving high rewards while still

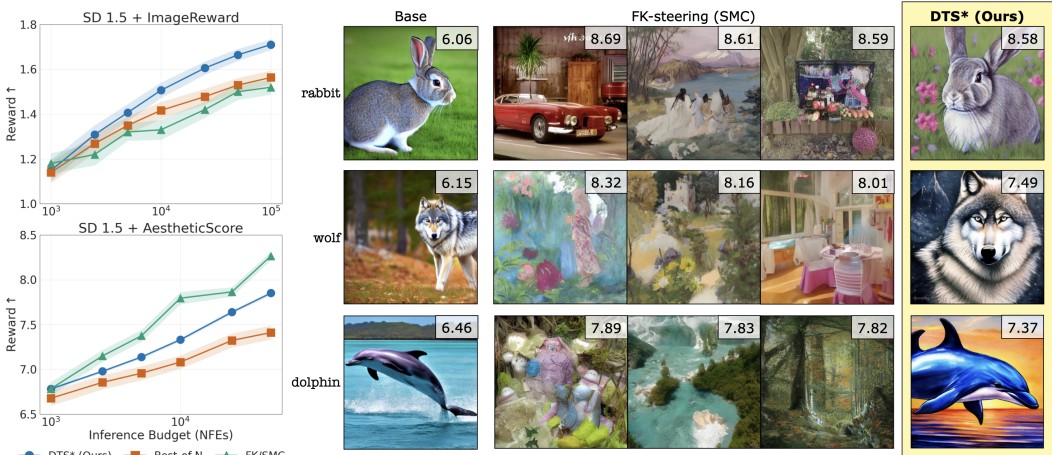

Figure 8: **Left:** Maximum ImageReward [88] vs. compute (NFEs) per prompt, averaged over 200 prompts from DrawBench [67], and maximum aesthetic score [69] vs. compute (NFEs) per prompt, averaged over 45 common animal prompts. **Right:** Samples generated using SD-v1.5 [64] for simple animal prompts and aesthetic score as the reward at $100k$ NFEs. For each prompt, $\text{DTS}^\star$ faithfully matches the prompt while achieving high reward, whereas SMC samples score higher but visibly over-optimize. Numbers in the corner show aesthetic scores.

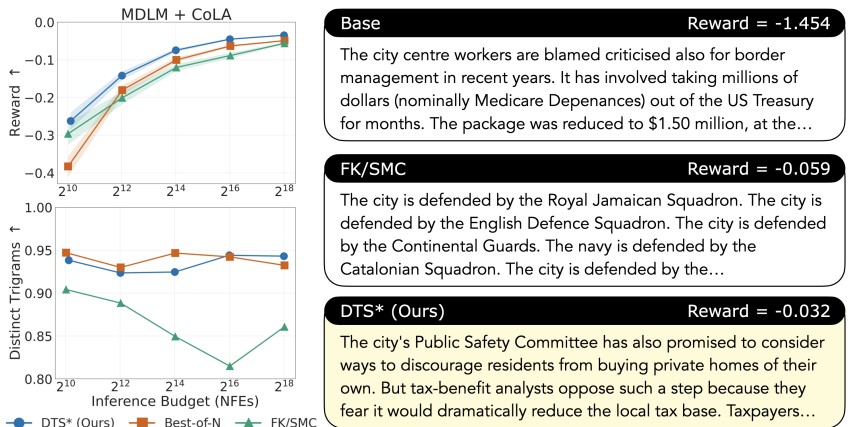

Figure 9: **Left:** Maximum reward and distinct trigrams vs. compute (NFEs) per prompt, across 15 simple prompts with MDLM [68] using a classifier trained on CoLA [54] as the reward. DTS$^\star$ obtains the highest reward while maintaining diverse outputs. **Right:** Typical samples generated by the base model, FK/SMC, and DTS$^\star$ for the prompt ''The city'' at $2^{18}$ NFEs.

maintaining faithfulness to the prior model. We hypothesize that since DTS$^\star$ backs up *soft* values, each node aggregates *posterior mass* rather than peak density. Consequently, a reward spike that lies in a vanishing-probability region of the prior contributes negligibly to the value estimates, resulting in an implicit KL-regularization effect (cf. Section 2).

## 5.3 Text generation

**Setting.** We evaluate DTS$^\star$ on text generation using MDLM [68], a discrete diffusion language model. We generate three text completions of length 64 for each of 15 prompts introduced by Han et al. [23]. The reward is defined as the log probability that the text is classified grammatically 'acceptable' by a BERT-based classifier [54] trained on the Corpus of Linguistic Acceptability (CoLA) [83]. We also report diversity by computing the number of distinct trigrams in each generated sequence. For decoding, we find that using DTS$^\star$ with max-backup ($\lambda \to \infty$) yields the best performance. We compare our method against two baselines: FK/SMC [71] and best-of-$N$.

**Results.** As shown in Figure 9, DTS$^\star$ consistently achieves the highest rewards as the number of function evaluations (NFEs) increases. Notably, reward functions in text domains are particularly susceptible to over-optimization [31; 25; 8], leading to the less diverse outputs observed when using FK/SMC. By contrast, DTS$^\star$ produces outputs that have both high rewards and high diversity. We perform evaluation using an LLM in Section J.4, since it aligns better with human judgement [47].

## 6 Discussion

We have introduced a novel framework that casts inference–time alignment of diffusion models as a finite-horizon tree search. By propagating terminal rewards via a soft value backup, our approach achieves global credit assignment and improved sample quality as compute increases. Below, we highlight practical considerations, point out limitations, and suggest directions for further work.

**High-dimensions and the role of pretrained model.** In high dimensions, an uninformed search tree grows exponentially with dimension, rendering pure tree search infeasible. A good quality pretrained model acts as a powerful prior, significantly pruning the effective search space. Even then, without gradient information, such methods can struggle if the posterior is very sharp, such as in certain inverse problems (Section I).

**Learning the value function.** In several applications of MCTS in game play, such as AlphaZero [70], deep neural networks approximate both policy and value. While our current work focuses on zero-shot inference-time alignment for *any* unseen reward, an exciting future direction would be to integrate a learned value network for a fixed reward.

**Compute cost.** The control flow of tree-based methods is sequential, which makes them less parallelizable than particle-based methods such as SMC. However, as discussed in Section 4.3, we implement an efficient version by batching calls to the diffusion model. Moreover, once the tree is constructed, sampling incurs no further model calls, making repeated draws effectively free.

## Acknowledgments and Disclosure of Funding

This research is in part supported by CIFAR AI Chairs and the NSERC Discovery program. Mila and the Digital Research Alliance of Canada provided computational resources.

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

# A  Extended related work

We discussed the main approaches that have been proposed for inference-time alignment of diffusion models in Section 3. Below, we briefly review additional steering works and three tangentially related areas: fine-tuning of diffusion models, their use in reinforcement learning, and entropy-regularized variations of Monte Carlo Tree Search.

**Additional steering methods.**  Some gradient-steering works treat guidance as stochastic control: Pandey et al. [56] learn KL-regularised drifts for differentiable rewards but break when high-reward modes lack coverage; Rout et al. [66] propose a training-free method to modify the drift for style transfer in vision; Huang et al. [33] extend to non-differentiable music rules via high-variance REINFORCE. Our search-based alignment does not require differentiability or dense coverage assumptions and is domain-agnostic. A different line of work focuses on plug-and-play samplers [86; 13] which also use Tweedie's formula to get biased values and use compute-heavy MCMC/Gibbs chains for inverse problems. Skreta et al. Our principled search method produces high-quality samples by using unbiased values and works with arbitrary rewards.

**Fine-tuning of diffusion models.**  To sample from the target distribution $\pi^*$ for a fixed, known reward function, one option is to amortize the posterior sampling problem and update the model parameters via fine-tuning. The paradigm mirrors the trajectory of large–language-model alignment [94; 61]. Supervised preference finetuning trains directly on synthetic pairs scored by a reward model [44; 85]. Some early methods exploit differentiable objectives to back-propagate a single scalar all the way to the noise prediction network [12; 58], whereas more traditional reinforcement learning approaches cast each reverse step as an action and optimize expected reward [5]. To avoid over-optimization of the reward, recent works use KL regularization [21; 78; 81].

**Diffusion models in reinforcement learning.**  Since the introduction of diffusion models as powerful frameworks for generative modeling, they have become popular for sampling actions or future states in RL. The earliest successes were in offline imitation learning, where some approaches model trajectories [36; 1] or expert policies [10] from offline datasets. Other works maximize a Q-function in addition to behavior cloning [82; 38], employ an explicit actor–critic scheme [24], or treat the critic as an energy function to guide the denoiser [49]. Some goal-conditioned extensions have also been proposed [62; 34]. Recent works have explored similar ideas in the online setting [90; 59; 35]. Those methods aim to maximize return for control tasks, while we aim to draw unbiased samples from the reward-tilted distribution for any chosen reward.

**Entropy-regularized MCTS.**  Monte-Carlo Tree-Search (MCTS) has recently been extended to soft-value objectives that incorporate an entropy bonus [87], which uses a log-sum-exp value update and samples actions from a Boltzmann distribution, guaranteeing improved exploration at the cost of converging to the soft rather than the standard optimum. Follow-up work proposed to adapt the entropy term to a predefined value [42] and decay the entropy term [55]. Very recently, Morozov et al. [53] used soft-backup MCTS to improve planning in Generative Flow Networks [4]. Our Diffusion Tree Sampling (DTS) follows the same Boltzmann selection and soft value backup pattern, it is the first to embed a *pre-trained diffusion kernel* inside the tree and to prove consistency for sampling from the KL-regularised posterior, not just selecting a single high-reward action. In this sense, DTS bridges the gap between entropy-regularized MCTS used for control and unbiased posterior sampling required for inference-time alignment of generative models.

# B  Sequential Monte Carlo for diffusion sampling

Many existing methods for inference-time diffusion alignment [84; 77; 7; 19; 39] apply sequential Monte Carlo (SMC) [15] to the reverse diffusion chain. SMC maintains a population of $K$ particles to approximately sample from a sequence of intermediate targets $\{\pi_t(\mathbf{x}_{t:T})\}_{t=T}^0$, culminating in the desired $\pi^*(\mathbf{x}_0) \propto p_\theta(\mathbf{x}_0) \exp(\lambda r(\mathbf{x}_0))$. In diffusion alignment, one usually sets

$$\pi_t(\mathbf{x}_{t:T}) \propto p(\mathbf{x}_T) \prod_{s=t+1}^{T} p_\theta(\mathbf{x}_{s-1} \mid \mathbf{x}_s) \exp\left(\lambda\,\hat{v}_t(\mathbf{x}_t)\right), \tag{10}$$

where $\hat{v}_t$ is a *potential* approximating the soft value $V_t$. Each SMC iteration for $t = T, T-1, \ldots, 0$ has three steps:

1. **Propagation.** Sample particles $\tilde{\mathbf{x}}_{t-1}^{(k)} \sim q_t(\cdot \mid \mathbf{x}_t^{(k)})$, for $k = 1, \ldots, K$ where $q_t$ is the proposal distribution, often set to be the diffusion transition $p_\theta(\cdot \mid \mathbf{x}_t)$.

2. **Weighting.** Assign importance weights

$$w_{t-1}^{(k)} = \underbrace{\frac{p_\theta(\tilde{\mathbf{x}}_{t-1}^{(k)} \mid \mathbf{x}_t^{(k)})}{q_t(\tilde{\mathbf{x}}_{t-1}^{(k)} \mid \mathbf{x}_t^{(k)})}}_{\text{importance ratio}} \times \exp\left(\lambda\, \hat{v}_{t-1}(\tilde{\mathbf{x}}_{t-1}^{(k)})\right). \tag{11}$$

The first factor corrects for using a proposal and the second tilts weights toward high estimated value.

3. **Resampling.** Resample $\{\tilde{\mathbf{x}}_{t-1}^{(k)}\}_{k=1}^K$ proportional to $\{w_{t-1}^{(k)}\}_{k=1}^K$ to obtain an equally weighted particle set $\{\mathbf{x}_{t-1}^{(k)}\}_{k=1}^K$ for the next iteration.

Classical SMC guarantees that, as $K \to \infty$ and if the potentials are exact, the empirical measure $\sum_k w_0^{(k)} \delta\left(\mathbf{x}_0^{(k)}\right)$ converges to the target distribution $\pi^*$, where $\delta(x)$ is the Dirac delta at $x$. In practice, however, this repeated sampling procedure can reduce the diversity of samples, especially when the weights have high variance. This results in an *effective sample size* which is much lower than $K$.

Another major issue when applying SMC to diffusion models is that estimating the soft value function $V_t$ is not straightforward and errors in the approximation can lead to inaccurate sampling. The next subsection discusses the *value-estimation problems* in more detail.

## C    Connection with Generative Flow Networks

Diffusion Tree Sampling can be viewed as an on-the-fly, non-parametric realization of the ideas behind Generative Flow Network (GFlowNet) [4]. Both frameworks ultimately seek to sample from an unnormalised density:

$$\pi^*(x) = \frac{1}{\mathcal{Z}}\, f(x), \quad \mathcal{Z} = \int f(x)\, dx,$$

but they do so with different machinery and at different points in the learning–inference pipeline.

GFlowNets assume a unique initial state $\mathbf{s}_0$ and define a probability over complete paths $\tau = (\mathbf{s}_0 \to \cdots \to \mathbf{s}_T = x)$ through

$$P_\theta(\tau) = \prod_{t=1}^T P_\theta(\mathbf{s}_t \mid \mathbf{s}_{t-1}) = P_\theta(\mathbf{s}_T) \prod_{t=1}^T P_B(\mathbf{s}_{t-1} \mid \mathbf{s}_t),$$

and train the parameters $\theta$ so that the *forward flow*, leaving every non-terminal state equals the *backward flow*, entering it plus injected terminal reward $r(\mathbf{x}) = \log f(\mathbf{x})$. For the special case of a tree-structured graph, this constraint in log form is a soft Bellman equation [76; 16; 52]:

$$F(\mathbf{s}) = \frac{1}{\lambda} \log \sum_{s' \in \text{Child}(s)} P_B(\mathbf{s} \mid \mathbf{s}') \exp\left(\lambda F(\mathbf{s}')\right),$$

with $F(\mathbf{s})$ the learned log-flow function.

DTS satisfies the same soft Bellman recursion (cf. Equation (5)), but does so *without* learning parameters. During tree construction, DTS estimates the soft value $V_t$ by Monte-Carlo log-sum-exp backups; selection then samples children proportionally to $\exp(\lambda \hat{v}_{t-1})$, where $\hat{v}_{t-1}$ is the estimated soft value. Repeated roll-outs make the empirical terminal distribution converge to the reward-tilted posterior $\pi^*(\mathbf{x}_0) \propto p_\theta(\mathbf{x}_0) \exp\left(\lambda r(\mathbf{x}_0)\right)$, just as a perfectly trained GFlowNet would.

The key differences between DTS and GFlowNets are summarized below.

- **Proposal.** DTS uses a *fixed*, pretrained diffusion kernel $p_\theta$ as a proposal, whereas GFlowNets learn their forward policy $P_\theta$.
- **Learning vs. search.** DTS performs pure inference without updating any parameters, whereas GFlowNets learn the parameters of the sampler to amortize future sampling.
- **Computational regime.** DTS excels when one has a strong prior and large *inference* budget for new rewards; GFlowNets shine when the reward is fixed and repeated queries amortize the *training* cost.

Because DTS is a search procedure, it is ideal for adapting a pretrained diffusion model to different unseen reward functions without retraining. GFlowNets, in contrast, learn a fast parametric sampler for a single reward.

## D Proofs and derivations

### D.1 Derivation of Equations 5 and 6

We derive the recursive relation satisfied by the soft value function as well as the expression for the optimal policy in Section 2 for completeness.

**Soft value function.** This recursive relation is analogous to the soft Bellman equation in maximum entropy RL [93; 22]. Starting from the definition of $V_t(\mathbf{x}_t)$:

$$
\begin{aligned}
V_t(\mathbf{x}_t) &= \frac{1}{\lambda} \log \mathbb{E}_{p_\theta(\mathbf{x}_{0:t-1}|\mathbf{x}_t)} \left[ \exp\left(\lambda r(\mathbf{x}_0)\right) \right] \\
&= \frac{1}{\lambda} \log \int p(\mathbf{x}_0, \mathbf{x}_1, \ldots, \mathbf{x}_{t-1}|\mathbf{x}_t) \exp\left(\lambda r(\mathbf{x}_0)\right) d\mathbf{x}_0 d\mathbf{x}_1 \ldots d\mathbf{x}_{t-1} \\
&= \frac{1}{\lambda} \log \int p(\mathbf{x}_0, \mathbf{x}_1, \ldots \mathbf{x}_{t-2}|\mathbf{x}_{t-1}) \, p(\mathbf{x}_{t-1}|\mathbf{x}_t) \exp\left(\lambda r(\mathbf{x}_0)\right) d\mathbf{x}_0 d\mathbf{x}_1 \ldots d\mathbf{x}_{t-1} \\
&= \frac{1}{\lambda} \log \int p(\mathbf{x}_{t-1}|\mathbf{x}_t) \underbrace{\left( \int p(\mathbf{x}_0, \mathbf{x}_1, \ldots \mathbf{x}_{t-2}|\mathbf{x}_{t-1}) \exp\left(\lambda r(\mathbf{x}_0)\right) d\mathbf{x}_0 d\mathbf{x}_1 \ldots d\mathbf{x}_{t-2} \right)}_{=\exp(\lambda V_{t-1}(\mathbf{x}_{t-1}))} d\mathbf{x}_{t-1} \\
&= \frac{1}{\lambda} \log \int p(\mathbf{x}_{t-1}|\mathbf{x}_t) \exp\left(\lambda V_{t-1}(\mathbf{x}_{t-1})\right) d\mathbf{x}_{t-1} = \frac{1}{\lambda} \log \mathbb{E}_{p(\mathbf{x}_{t-1}|\mathbf{x}_t)} \left[ \exp\left(\lambda V_{t-1}(\mathbf{x}_{t-1})\right) \right].
\end{aligned}
$$

The above relation combined with the terminal condition $V_0(\mathbf{x}_0) = r(\mathbf{x}_0)$ gives Equation (5).

**Optimal policy.** The joint target density over the full chain $(\mathbf{x}_0, \ldots, \mathbf{x}_{t-1}, \mathbf{x}_t)$ is given by:

$$
\pi^*(\mathbf{x}_0, \ldots, \mathbf{x}_{t-1}, \mathbf{x}_t) = \frac{1}{\mathcal{Z}} \, p_\theta(\mathbf{x}_0, \ldots, \mathbf{x}_{t-1}, \mathbf{x}_t) \, \exp\left(\lambda r(\mathbf{x}_0)\right),
$$

where $\mathcal{Z}$ represent the normalization constant of this joint density.

The marginal joint density of $(\mathbf{x}_t, \mathbf{x}_{t-1})$ under $\pi^*$ is:

$$
\begin{aligned}
\pi^*(\mathbf{x}_t, \mathbf{x}_{t-1}) &= \frac{1}{\mathcal{Z}} \int p_\theta(\mathbf{x}_0, \ldots, \mathbf{x}_{t-1}, \mathbf{x}_t) \exp\left(\lambda r(\mathbf{x}_0)\right) d\mathbf{x}_0 \ldots d\mathbf{x}_{t-2} \\
&= \frac{1}{\mathcal{Z}} p_\theta(\mathbf{x}_t) p_\theta(\mathbf{x}_{t-1} \mid \mathbf{x}_t) \left( \int p_\theta(\mathbf{x}_0, \ldots, \mathbf{x}_{t-2} \mid \mathbf{x}_{t-1}) \exp\left(\lambda r(\mathbf{x}_0)\right) d\mathbf{x}_0 \ldots d\mathbf{x}_{t-2} \right) \\
&= \frac{1}{\mathcal{Z}} p_\theta(\mathbf{x}_t) p_\theta(\mathbf{x}_{t-1} \mid \mathbf{x}_t) \exp\left(\lambda V_{t-1}(\mathbf{x}_{t-1})\right)
\end{aligned}
$$

Similarly, the marginal density of $\mathbf{x}_t$ under $\pi^*$ is:

$$
\pi^*(\mathbf{x}_t) = \frac{1}{\mathcal{Z}} \, p_\theta(\mathbf{x}_t) \exp\left(\lambda V_t(\mathbf{x}_t)\right)
$$

By dividing these two marginals, we get the transitions under the optimal policy:

$$
\begin{aligned}
\pi^*(\mathbf{x}_{t-1} \mid \mathbf{x}_t) = \frac{\pi^*(\mathbf{x}_t, \mathbf{x}_{t-1})}{\pi^*(\mathbf{x}_t)} &= \frac{p_\theta(\mathbf{x}_{t-1} \mid \mathbf{x}_t) \exp\left(\lambda V_{t-1}(\mathbf{x}_{t-1})\right)}{\exp\left(\lambda V_t(\mathbf{x}_t)\right)} \\
&= \frac{p_\theta(\mathbf{x}_{t-1} \mid \mathbf{x}_t) \exp\left(\lambda V_{t-1}(\mathbf{x}_{t-1})\right)}{\int p_\theta(\mathbf{x}_{t-1} \mid \mathbf{x}_t) \exp\left(\lambda V_{t-1}(\mathbf{x}_{t-1})\right) d\mathbf{x}_{t-1}}.
\end{aligned}
$$

The above relation gives the optimal policy from Equation (6).

### D.2 Proof of Proposition 1

**Proposition 1** (Asymptotic consistency). *Let $r$ be bounded and $\lambda > 0$, then* DTS *produces a sequence of terminal states whose empirical distribution converges to the optimal policy $\pi^*$ as the number of tree iterations $M \to \infty$.*

*Proof.* We use $p(\cdot \mid \mathbf{x}_t)$ to denote a general proposal distribution. For application to diffusion alignment, this would correspond to transitions under the pretrained model $p_\theta(\cdot \mid \mathbf{x}_t)$. Additionally, we use $\hat{q}(\cdot \mid \mathbf{x}_t)$ to denote the transition density of DTS.

**Step 1: Transition probability under DTS.** Recall that under DTS, given a node $\mathbf{x}_t$, we create each child by sampling from the base model $p(\cdot \mid \mathbf{x}_t)$. During tree traversal, we select the next state $\mathbf{x}_{t-1}$ proportional to the exponentiated soft value function. Thus, the transition probability of DTS from $\mathbf{x}_t$ to $\mathbf{x}_{t-1}$ is given by:

$$\hat{q}(\mathbf{x}_{t-1} \mid \mathbf{x}_t) = \frac{p(\mathbf{x}_{t-1} \mid \mathbf{x}_t) \exp\left(\lambda \hat{v}(\mathbf{x}_{t-1})\right)}{\int p(\mathbf{x}_{t-1} \mid \mathbf{x}_t) \exp\left(\lambda \hat{v}(\mathbf{x}_{t-1})\right) \, d\mathbf{x}_{t-1}} = \frac{p(\mathbf{x}_{t-1} \mid \mathbf{x}_t) \exp\left(\lambda \hat{v}(\mathbf{x}_{t-1})\right)}{\exp\left(\lambda \hat{v}(\mathbf{x}_t)\right)}, \quad (12)$$

where the second equality follows from the definition of the soft Bellman equation:

$$\hat{v}(\mathbf{x}_t) = \frac{1}{\lambda} \log \mathbb{E}_{\mathbf{x}_{t-1} \sim p(\cdot \mid \mathbf{x}_t)} \left[\exp\left(\lambda \hat{v}(\mathbf{x}_{t-1})\right)\right].$$

**Step 2: Joint density of trajectory.** Recall that the root node of DTS contains a dummy state $\mathbf{x}_{T+1}$ that transitions to the diffusion process prior $\hat{q}(\mathbf{x}_T \mid \mathbf{x}_{T+1}) = \mathcal{N}(0, I)$. Then, the joint density of a full trajectory $\{\mathbf{x}_T, \mathbf{x}_{T-1}, \ldots, \mathbf{x}_0\}$ under DTS is given by:

$$\begin{aligned}
\hat{q}(\mathbf{x}_T, \mathbf{x}_{T-1}, \ldots, \mathbf{x}_0) &= \prod_{t=1}^{T+1} \hat{q}(\mathbf{x}_{t-1} \mid \mathbf{x}_t) = \prod_{t=1}^{T+1} \frac{p(\mathbf{x}_{t-1} \mid \mathbf{x}_t) \exp\left(\lambda \hat{v}(\mathbf{x}_{t-1})\right)}{\exp\left(\lambda \hat{v}(\mathbf{x}_t)\right)} \\
&= \frac{\exp\left(\lambda \hat{v}(\mathbf{x}_0)\right)}{\exp\left(\lambda \hat{v}(\mathbf{x}_{T+1})\right)} \prod_{t=1}^{T+1} p(\mathbf{x}_{t-1} \mid \mathbf{x}_t) \\
&= \frac{\exp\left(\lambda \hat{v}(\mathbf{x}_0)\right)}{\exp\left(\lambda \hat{v}(\mathbf{x}_{T+1})\right)} p(\mathbf{x}_T, \mathbf{x}_{T-1}, \ldots, \mathbf{x}_0).
\end{aligned}$$

**Step 3: Marginalizing.** Marginalizing over intermediate states $\mathbf{x}_1, \ldots, \mathbf{x}_T$, we get the distribution of terminal state $\mathbf{x}_0$:

$$\begin{aligned}
\hat{q}(\mathbf{x}_0) &= \int \hat{q}(\mathbf{x}_T, \mathbf{x}_{T-1}, \ldots, \mathbf{x}_0) \, d\mathbf{x}_T d\mathbf{x}_{T-1} \ldots d\mathbf{x}_1 \\
&= \frac{\exp\left(\lambda \hat{v}(\mathbf{x}_0)\right)}{\exp\left(\lambda \hat{v}(\mathbf{x}_{T+1})\right)} \int p(\mathbf{x}_T, \mathbf{x}_{T-1}, \ldots, \mathbf{x}_0) \, d\mathbf{x}_T d\mathbf{x}_{T-1} \ldots d\mathbf{x}_1 \\
&= \frac{\exp\left(\lambda \hat{v}(\mathbf{x}_0)\right)}{\exp\left(\lambda \hat{v}(\mathbf{x}_{T+1})\right)} p(\mathbf{x}_0).
\end{aligned}$$

By definition, the soft value function at the terminal node is $\hat{v}(\mathbf{x}_0) = r(\mathbf{x}_0)$. Plugging this and using the definition of value function from Equation (4), we have:

$$\begin{aligned}
\hat{q}(\mathbf{x}_0) &= \frac{\exp\left(\lambda r(\mathbf{x}_0)\right) p(\mathbf{x}_0)}{\int p(\mathbf{x}_T, \mathbf{x}_{T-1}, \ldots, \mathbf{x}_0 \mid \mathbf{x}_{T+1}) \exp\left(\lambda r(\mathbf{x}_0)\right) \, d\mathbf{x}_T d\mathbf{x}_{T-1} \ldots d\mathbf{x}_1 d\mathbf{x}_0} \\
&= \frac{\exp\left(\lambda r(\mathbf{x}_0)\right) p(\mathbf{x}_0)}{\int p(\mathbf{x}_0) \exp\left(\lambda r(\mathbf{x}_0)\right) \, d\mathbf{x}_0}.
\end{aligned}$$

This has the form of the target distribution in Equation (6), except that it uses the value estimates $\hat{v}$ that are calculated based on rollouts starting from each state $\mathbf{x}_t$. In the limit of infinite rollouts, these value estimates approach the true soft values, confirming that the sampling distribution $\hat{q}$ from DTS exactly matches the target distribution $\pi^*$.

Therefore, DTS is consistent, as it correctly generates samples from the desired target distribution asymptotically.

$\square$

# E DTS and DTS$^\star$ algorithm

---

**Algorithm 1** Diffusion Tree Sampling (DTS) and Diffusion Tree Search (DTS$^\star$)

---

1: **Input:** base policy $p_\theta$, reward function $r$, number of iterations $M$, inverse temperature $\lambda$, parameters $C, \alpha, c_{\text{uct}}$
2: Initialize root node $\mathbf{x}_{T+1}$ with dummy value, $\hat{v}(\mathbf{x}_{T+1}) = 0$, $N(\mathbf{x}_{T+1}) = 1$
3: Initialize tree $\mathcal{T}$ with root node $\mathbf{x}_{T+1}$
4: **for** $m = 1, \ldots, M$ **do**
5:     $P \leftarrow \{\mathbf{x}_{T+1}\}$
6:     Set $t \leftarrow T + 1$
7:     // Selection
8:     **while** $|\mathcal{C}(\mathbf{x}_t)| \geq C \cdot N(\mathbf{x}_t)^\alpha$ and $t > 0$ **do**
9:         **[DTS]** select child probabilistically: $\mathbf{x}_{t-1} \sim \frac{\exp(\lambda \hat{v}(\mathbf{x}_{t-1}))}{\sum_{\mathbf{x}' \in \mathcal{C}(\mathbf{x}_t)} \exp(\lambda \hat{v}(\mathbf{x}'))}$
10:         **[DTS$^\star$]** select child maximizing UCT: $\mathbf{x}_{t-1} = \arg\max_{\mathbf{x}' \in \mathcal{C}(\mathbf{x}_t)} \hat{v}(\mathbf{x}') + c_{\text{uct}} \sqrt{\frac{\log N(\mathbf{x}_t)}{N(\mathbf{x}')}}$
11:         $P \leftarrow P \cup \{\mathbf{x}_{t-1}\}$
12:         $t \leftarrow t - 1$
13:     **end while**
14:     // Expansion: expand $\mathbf{x}_t$ by sampling a new child
15:     **if** $t > 0$, and $|\mathcal{C}(\mathbf{x}_t)| < C \cdot N(\mathbf{x}_t)^\alpha$ **then**
16:         // Rollout: from new node $\mathbf{x}_{t-1}$ sample rollout path to terminal $\mathbf{x}_0$
17:         **while** $t > 0$ **do**
18:             $\mathbf{x}_{t-1} \sim p_\theta(\cdot \mid \mathbf{x}_t), \quad \hat{v}(\mathbf{x}_{t-1}) = 0, \quad N(\mathbf{x}_{t-1}) = 1$
19:             $P \leftarrow P \cup \{\mathbf{x}_{t-1}\}$
20:             $t \leftarrow t - 1$
21:         **end while**
22:     **end if**
23:     Evaluate terminal reward: $\hat{v}(\mathbf{x}_0) = r(\mathbf{x}_0)$
24:     // Backup: update value along path $P$
25:     **for** $t = 0, \ldots, T$ **do**
26:         Soft backup: $\hat{v}(\mathbf{x}_{t+1}) \leftarrow \frac{1}{\lambda} \log \left( \frac{1}{|\mathcal{C}(\mathbf{x}_{t+1})|} \sum_{\mathbf{x}_t \in \mathcal{C}(\mathbf{x}_{t+1})} \exp(\lambda \hat{v}(\mathbf{x}_t)) \right)$
27:         Update visits: $N(\mathbf{x}_{t+1}) \leftarrow N(\mathbf{x}_{t+1}) + 1$
28:     **end for**
29: **end for**
30: **return** $\mathcal{T}$

---

**Algorithm 2** Diffusion Tree Sampling (DTS) and Diffusion Tree Search (DTS$^\star$) inference

---

1: **Input:** Constructed tree $\mathcal{T}$, number of samples $N$, inverse temperature $\lambda$
2: Initialize population of samples $\mathcal{S} \leftarrow \phi$
3: **for** $n = 1, \ldots, N$ **do**
4:     Set $t \leftarrow T + 1$
5:     **while** $t > 0$ **do**
6:         **[DTS]** select child probabilistically: $\mathbf{x}_{t-1} \sim \frac{\exp(\lambda \hat{v}(\mathbf{x}_{t-1}))}{\sum_{\mathbf{x}' \in \mathcal{C}(\mathbf{x}_t)} \exp(\lambda \hat{v}(\mathbf{x}'))}$
7:         **[DTS$^\star$]** select child maximizing value: $\mathbf{x}_{t-1} = \arg\max_{\mathbf{x}' \in \mathcal{C}(\mathbf{x}_t)} \hat{v}(\mathbf{x}')$
8:         $t \leftarrow t - 1$
9:     **end while**
10:     $\mathcal{S} \leftarrow \mathcal{S} \cup \{\mathbf{x}_0\}$
11: **end for**
12: **return** Samples $\mathcal{S}$

---

# F Implementation details for DTS and DTS$^\star$

## F.1 Tree structure

The algorithm presented in Section 4.2 and Section E allows every state $\mathbf{x}_t$ along the denoising trajectory to be considered for branching. However, in practice, we only branch every few timesteps.

We noticed very little difference in performance between the two cases for the same number of function evaluations, however, we expect branching at every step to outperform for a very high compute budget. We match the tree branching schedule with the resampling schedule for all baselines with SMC, similar to the setting from Singhal et al. [71]. The exact setting for each experiment is presented in Table 2, where we always branch at the root node corresponding to $t = T$.

Table 2: Branching schedule for DTS and DTS$^\star$, which is also the resampling schedule used for SMC-based methods – SMC/FK [71], TDS [84], DAS [39].

| Domain | Total denoising steps | Branching schedule |
|---|---|---|
| Two-dimensional | 100 | $100(\text{root}), 80, 60, 40, 20$ |
| Image pixels (MNIST, CIFAR-10) | 50 | $50(\text{root}), 40, 30, 20, 10$ |
| Image latents (SD-v1.5) | 50 | $50(\text{root}), 40, 30, 20, 10$ |
| Text tokens (MDLM) | 64 | $64(\text{root}), 54, 44, 34, 24, 14$ |

Apart from this, we have hyperparameters associated with progressive widening that control the maximum number of branches at any node. We used $\alpha = 0.8$ and $C = 2$ for all two-dimensional and image experiments and $\alpha = 0.7$ and $C = 2$ for text generation. There is a scope of improving the performance of DTS and DTS$^\star$ further by tuning these parameters for specific tasks.

## F.2 Experiment details

---

**Illustrative 2D**

**Base diffusion model:** The denoising network is an MLP that takes as input the 2-dimensional data $\mathbf{x}_t$ and the timestep $t$ and outputs a 2-dimensional noise prediction. The timestep is transformed using sinusoidal embeddings [80]. The network has four hidden layers of 128 dimension each with the sigmoid linear unit (SiLU, [28]) activation. We used the linear noise schedule with $\beta_{\min} = 0.001$ and $\beta_{\max} = 0.07$ and the score matching objective. The optimizer used for training was Adam [40] with a learning rate of $3 \times 10^{-3}$. We train the model for 500 epochs on a training set of 10000 samples.

**Reward function:** *Gaussian mixture:* The reward function is:

$$r(\mathbf{x}) = \log\left(\sum_{i=1}^{8} w_i \exp\left(-\|\mathbf{x} - \boldsymbol{\mu}_i\|^2/2\sigma^2\right)\right),$$

where $w_i = \exp(1.5\,i)$, $\quad \boldsymbol{\mu}_i = 4\left(\cos\frac{2\pi(i-1)}{8}, \sin\frac{2\pi(i-1)}{8}\right)$, $\quad i = 1, \ldots, 8$, with $\sigma = 0.3$.
*Checkerboard:* The reward is negative distance from the center $r(\mathbf{x}) = -0.5\|\mathbf{x}\|^2$.

---

**Class-conditional MNIST**

**Base diffusion model:** The denoising network is a Unet architecture [65] that operates on images of size $32 \times 32 \times 1$ (upscaled from $28 \times 28 \times 1$) with block channels $\{32, 64, 128, 256\}$. We use the DDIMScheduler$^a$ from diffusers library with default parameters, except we set $\eta = 1.0$ so the inference process is stochastic like DDPMs [30]. We use the AdamW optimizer with a learning rate of $10^{-4}$ for 100 epochs on the MNIST training set.

**Reward function:** We train a classifier $p(c \mid \mathbf{x})$ on the MNIST training set. The classifier is a convolutional neural network [43] using two $5 \times 5$ kernels with $(16, 32)$ channels followed by $2 \times 2$ max pooling operation with ReLU activations. The features are then flattened and followed by a linear layer with 10 outputs corresponding to the classes. The network was trained using Adam optimizer with learning rate $10^{-3}$. The reward function for single class generation is the log likelihood of the class $r_i(\mathbf{x}) = \log p(c = i \mid \mathbf{x})$ for $i \in \{0, 1, \ldots, 9\}$. For the even or odd generation, it is defined as $r(\mathbf{x}) = \max_{i \in \mathcal{S}} \log p(c = i \mid \mathbf{x})$, where $\mathcal{S} = \{0, 2, 4, 6, 8\}$ for even digit generation and $\mathcal{S} = \{1, 3, 5, 7, 9\}$ for odd digit generation.

---

$^a$https://huggingface.co/docs/diffusers/en/api/schedulers/ddim

### Class-conditional CIFAR-10

**Base diffusion model:** We used the pre-trained diffusion model `ddpm-cifar10-32`[a] from Hugging Face, which uses a Unet architecture and diffuses over $32 \times 32 \times 3$ images in pixel-space. We use the DDIMScheduler with $\eta = 1.0$ for stochastic denoising.

**Reward function:** We train a classifier $p(c \mid \mathbf{x})$ on the CIFAR-10 training set. The classifier uses a ResNet-18 [26] backbone that outputs an embedding which is average pooled, flattened, and passed to a single linear layer with 10 outputs. The network is trained using Adam optimizer with learning rate $10^{-3}$. Similar to MNIST single class generation, the reward function is the log likelihood of the class $r_i(\mathbf{x}) = \log p(c = i \mid \mathbf{x})$ for $i \in \{0, 1, \ldots, 9\}$.

_______________

[a]`https://huggingface.co/google/ddpm-cifar10-32`

---

### Text-to-image

**Base diffusion model:** We use Stable Diffusion v1.5[a] from Hugging Face, which is a latent diffusion model [64]. The diffusion process is defined over $64 \times 64 \times 4$ latent variables, which are obtained by encoding $512 \times 512 \times 3$ images using a variational autoencoder. The model uses CLIP [60] to encode text prompts into embeddings which are then used to condition the generative process via classifier-free guidance [29]. We use the DDIMScheduler with $\eta = 1.0$.

**Reward function:** We use pre-trained models as reward functions including ImageReward[b] $r(\mathbf{x}, \mathbf{y})$ that encodes prompt accuracy as well as human preferences the LAION aesthetic score predictor[c] $r(\mathbf{x})$ that encodes aesthetic quality of an image.

_______________

[a]`https://huggingface.co/stable-diffusion-v1-5/stable-diffusion-v1-5`
[b]`https://github.com/THUDM/ImageReward`
[c]`https://github.com/LAION-AI/aesthetic-predictor`

---

### Conditional text

**Base diffusion model:** We use MDLM[a] for our text generation experiments. This is a discrete diffusion model with 110M parameters that directly predicts the tokens. We define the diffusion process over a context length of 64 tokens with 64 sampling steps and use the standard discrete unmasking update for stochastic denoising.

**Reward function:** We use a BERT-based classifier[b] trained on the Corpus of Linguistic Acceptability (CoLA) [83]. This reward function $r(\mathbf{x})$ encodes the linguistic acceptability of a given string $\mathbf{x}$. The reward is the log probability of the text being "acceptable". We find this model to be more robust to reward-hacking than alternatives.

_______________

[a]`https://huggingface.co/kuleshov-group/mdlm-owt`
[b]`https://huggingface.co/textattack/roberta-base-CoLA`

---

### F.3 Compute

We report execution times on a single A100 GPU with 80 gigabytes of memory.

- Each 2D experiment including all methods runs in 15 minutes. Adding up the time over five seeds and two different datasets, the combined run time is approximately 2.5 GPU hours.
- The MNIST and CIFAR-10 class-conditional experiments use approximately 3 GPU hours per class including all methods. Over all 22 tasks (10 MNIST single digit + 2 MNIST even/odd + 10 CIFAR-10 classes) equals approximately 66 GPU hours.
- The text-to-image experiments using Stable Diffusion v1.5 require roughly 30 minutes per prompt across all methods. Adding up all 200 prompts from DrawBench and 45 animal prompts, reproducing all experiments requires approximately 123 GPU hours.
- The text generation experiments using MDLM requires roughly 30 minutes per prompt. Thus, generating 3 completions per prompt for the 15 prompts requires roughly 22.5 GPU hours.

# G  Details of baselines

We re-implemented all baseline methods in our unified codebase since most of them use SMC as a backbone and share the same underlying infrastructure. Each implementation was validated by reproducing the quantitative results reported in its original paper. Section B provides a concise primer on SMC for reference. The complete source code including all baselines will be released publicly upon publication of this work.

**DPS.**  Diffusion Posterior Sampling [11] was originally proposed for noisy inverse problems such as image super-resolution and de-blurring using the gradient of the final objective. To adapt this method for general reward functions, we make a minor modification by replacing the gradient of the inverse problem objective with the gradient of the reward function:

$$\tilde{\mathbf{x}}_{t-1} \sim \mathcal{N}\left(\mu_\theta(\mathbf{x}_t, t) + \lambda\, \sigma_t^2\, \nabla_{\mathbf{x}_t} r(\hat{\mathbf{x}}_0(\mathbf{x}_t)), \sigma_t^2\, I\right), \tag{13}$$

where $\hat{\mathbf{x}}_0$ is obtained using Tweedie's formula (cf. Section 3), $\mu_\theta$ is the predicted mean of the base diffusion model, and $r(\mathbf{x})$ is the reward function in the two-dimensional experiments and classifier log likelihoods $\log p(c = i \mid \mathbf{x})$ for class-conditional image experiments. The official implementation is provided here.

**SMC/FK-Steering.**  In our paper, SMC refers to the simplest variant, FK-Steering [71], which defines different weighting schemes and uses the pre-trained diffusion model as the proposal distribution. As per the setting in Singhal et al. [71], we perform the resampling step at fixed intervals during denoising (given in Table 2) and use adaptive resampling to increase diversity of generated samples. Our sampling experiments (two-dimensional and class-conditional image generation) use the 'diff' potential with $\lambda = 1.0$, whereas the search experiments (text-to-image and text generation) use the 'max' potential with $\lambda = 10.0$. The weights for resampling are given by Equation (11) where the proposal is equal to the pre-trained diffusion transition and the value estimates are equal to:

$$\hat{v}_{t-1}^{\text{diff}}(\tilde{\mathbf{x}}_{t-1}) = r\left(\hat{\mathbf{x}}_0(\tilde{\mathbf{x}}_{t-1})\right) - r\left(\hat{\mathbf{x}}_0(\mathbf{x}_t)\right), \quad \hat{v}_T^{\text{diff}}(\mathbf{x}_T) = r\left(\hat{\mathbf{x}}_0(\mathbf{x}_T)\right).$$
$$\hat{v}_{t-1}^{\text{max}}(\tilde{\mathbf{x}}_{t-1}) = \max\left\{r(\hat{\mathbf{x}}_0(\tilde{\mathbf{x}}_{t-1})), m_t^{(k)}\right\}, \quad m_t^{(k)} = \max_{s \geq t} r\left(\hat{\mathbf{x}}_0(\mathbf{x}_s^{(k)})\right).$$

We adapted the official implementation provided here.

**TDS.**  Twisted Diffusion Sampler [84] comprises of a "twisted" proposal which is used along with SMC to sample from the target posterior distribution. For general reward functions, the twisted proposal is the same as the one used in Equation (13) and the final weights are obtained using Equation (11) after plugging in the twisted proposal and the value estimates:

$$q_t(\tilde{\mathbf{x}}_{t-1} \mid \mathbf{x}_t) = \mathcal{N}\left(\tilde{\mathbf{x}}_{t-1}\,;\, \mu_\theta(\mathbf{x}_t, t) + \lambda\, \sigma_t^2\, \nabla_{\mathbf{x}_t} r(\hat{\mathbf{x}}_0(\mathbf{x}_t)), \sigma_t^2\, I\right).$$
$$\hat{v}_{t-1}(\tilde{\mathbf{x}}_{t-1}) = r\left(\hat{\mathbf{x}}_0(\tilde{\mathbf{x}}_{t-1})\right) - r\left(\hat{\mathbf{x}}_0(\mathbf{x}_t)\right), \quad \hat{v}_T(\mathbf{x}_T) = r\left(\hat{\mathbf{x}}_0(\mathbf{x}_T)\right).$$

The official implementation is provided here.

**DAS.**  Diffusion Alignment as Sampling [39] re-uses the twisted proposal of TDS but multiplies the reward term by a monotone tempering schedule $0 = \gamma_T \leq \gamma_{T-1} \leq \ldots \leq \gamma_0 = 1$ to reduce the bias from inaccurate value estimates at high noise levels. The weights are given by Equation (11) after plugging in the tempered proposal and value estimates:

$$q_t(\tilde{\mathbf{x}}_{t-1} \mid \mathbf{x}_t) = \mathcal{N}\left(\tilde{\mathbf{x}}_{t-1}\,;\, \mu_\theta(\mathbf{x}_t, t) + \lambda\, \gamma_{t-1}\, \sigma_t^2\, \nabla_{\mathbf{x}_t} r(\hat{\mathbf{x}}_0(\mathbf{x}_t)), \sigma_t^2\, I\right).$$
$$\hat{v}_{t-1}(\tilde{\mathbf{x}}_{t-1}) = \gamma_{t-1}\, r\left(\hat{\mathbf{x}}_0(\tilde{\mathbf{x}}_{t-1})\right) - \gamma_t\, r\left(\hat{\mathbf{x}}_0(\mathbf{x}_t)\right), \quad \hat{v}_T(\mathbf{x}_T) = \gamma_T\, r\left(\hat{\mathbf{x}}_0(\mathbf{x}_T)\right).$$

The official implementation is provided here.

# H  Memory and wall-clock time comparison

Below, we provide peak memory usage for each method in the text-to-image generation experiment using Stable Diffusion 1.5 and ImageReward. We perform this experiment with 40 different prompts and average the results. The numbers are reported for a single NVIDIA A100 GPU with 40GB of memory using the PyTorch profiler.

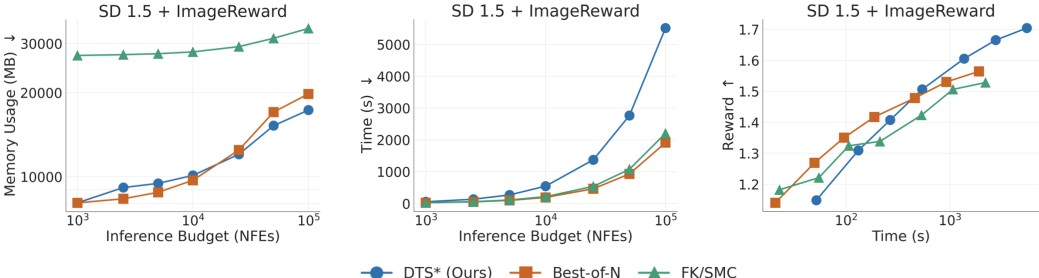

Figure 10: Comparison of methods using SD-v1.5 and ImageReward. **Left:** Peak memory usage (MB) for different NFEs. DTS$^\star$ is comparable to best-of-N, whereas SMC/FK requires much more memory. **Center:** Wall-clock time (seconds) for different NFEs. **Right:** Reward vs. wall-clock time, showing that DTS$^\star$ scales more favorably in terms of runtime when accounting for its performance. At higher levels of compute, DTS$^\star$ can match the performance of best-of-N with roughly 2× less wall-clock time. The wall-clock time for DTS$^\star$ can be greatly reduced by using larger batch sizes, at the expense of slightly more memory and possibly a slight degradation in performance. We used a batch size of 1 here to keep in line with the rest of the experiments.

**Peak Memory Usage.** We observe that the memory usage of DTS$^\star$ is much lower compared to FK/SMC, and is more comparable to best-of-N. Initially, DTS$^\star$ uses slightly more memory than best-of-N, but with more compute, it tilts in favor of DTS$^\star$, showing that it actually scales better in terms of memory.

The main memory usage of DTS$^\star$ is the tree as noted by the reviewer, which stores $64 \times 64 \times 4$ latents. Note that currently the entire tree is kept on the GPU, and we can save GPU memory by offloading the tree to the CPU. For best-of-N, the memory usage peaks at the end when all $N$ candidates need to be decoded using the VAE into $512 \times 512 \times 3$ images and evaluated using the reward function to pick the best sample. This problem does not occur with DTS$^\star$, since the rewards are calculated when traversal reaches the terminal state, and multiple high-resolution images do not need to be kept in memory. For FK/SMC, this cost is very high since all N particles need to be decoded into $512 \times 512 \times 3$ images, then evaluated using the reward function at *each resampling step*. In general, for images and text, the reward function itself is a large pretrained model, and computing the reward requires a non-trivial amount of compute.

**Wall-clock time.** In terms of wall clock time, DTS$^\star$ is approximately 2.5× slower than the other methods for the same number of function evaluations (NFEs), since it is harder to parallelize. Note that the runtime reflects tree building from scratch – once the tree is built, repeated sampling is just pointer chasing and is near instantaneous (whereas for other methods repeated sampling would require starting the entire procedure again).

We discuss two important considerations regarding the runtime:

- When taking into account their performance, DTS$^\star$ is still preferable in terms of wall clock time since it outperforms the baselines and needs much fewer NFEs to reach the same level of performance. As noted in the paper and the table below, for the text-to-image task, it needs roughly 5× fewer NFEs to match the reward of the best performing baseline (best-of-N) at $100k$ NFEs. In other words, to match the performance of best-of-N at $100k$ NFEs, DTS$^\star$ is approximately 2× faster in terms of wall clock time.

- We can significantly reduce the runtime of DTS/DTS$^\star$ by using multiple actors in parallel that traverse and update a shared tree (similar to AlphaZero [70] training). This will give a roughly linear speedup in runtime at the cost of increased memory for copying the diffusion model (the tree is not copied since it can be shared among all actors, hence increase in memory will be sublinear in terms of the number of actors). This represents an inherent trade-off between memory usage and runtime. We did not implement a distributed pipeline since it requires some engineering effort, but it is a viable option.

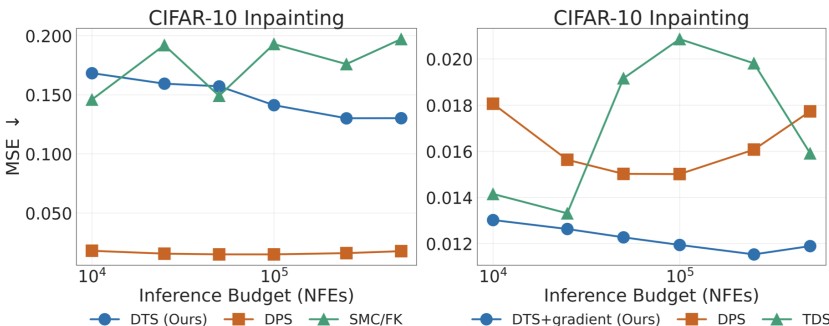

Figure 11: Mean square error (MSE) between generated images and reference image on 20 randomly sampled images from CIFAR-10. Methods that do not use gradient information (SMC, DTS) during inference struggle due to the very sharp posterior. Incorporating the gradient into DTS allows it to outperform baselines and continue improving with more compute.

# I   Image inpainting experiments

As an added application, we test the feasibility of our method on solving inverse problems. We run an inpainting experiment on CIFAR-10 images with a Gaussian forward channel as in Chung et al. [11]. We run this experiment on 20 images (two images sampled randomly from each class) and average the results. For each $32 \times 32$ image, we mask the $10 \times 10$ patch from the center and use the negative error between the generated sample and the unmasked pixels from the original image as the reward. We report the final average mean squared error (MSE) between 16 generated images and the reference image.

Figure 11 shows that methods that do not use gradient information, such as SMC [71] and DTS, do not seem to perform well on this task. We hypothesize this is because the inpainting task has a very sharp posterior where the sampled image must exactly match the unmasked pixels (compared to class conditional sampling where the posterior represents a semantic concept and has a wider mode). This is a very hard search problem, and without the forward channel gradient to guide the sampling process, there is very little signal in most regions of the pretrained model distribution.

We then incorporate the gradient into DTS, where we use the gradient term in addition to the score model to obtain the next state when constructing the tree. The backup and selection procedure remains the same. With the gradient-guided proposal, DTS improves upon DPS [11] and even SMC + gradient (which is the same as TDS [84]). We also observe that DTS allows the performance to consistently improve with more compute, whereas vanilla DPS does not improve with more compute. At $500k$ NFEs, DTS + gradient improves $32.96\%$ compared to vanilla DPS.

# J   Additional experimental results

## J.1   Comparison between DTS and DTS$^\star$

We compare DTS and DTS$^\star$ on the text-to-image setting using Stable Diffusion 1.5 with ImageReward on 40 prompts from DrawBench. We picked complex prompts where the base model struggles to generate high reward images to demonstrate the difference more clearly. For DTS, we sample 16 images from the tree and pick the one with the highest reward. The results show that there is a noticeable gap, since DTS spends more compute expanding suboptimal paths, whereas DTS$^\star$ aggressively searches only the highest reward regions except for the small exploration bonus.

Table 3: DTS and DTS$^\star$ with SDv-1.5 and ImageReward on 40 prompts from DrawBench.

| Algorithm $\downarrow$ NFEs $\rightarrow$ | 1000 | 2500 | 5000 | 10000 | 25000 | 50000 |
|---|---|---|---|---|---|---|
| DTS | 0.774 | 1.025 | 1.064 | 1.112 | 1.214 | 1.238 |
| DTS$^\star$ | 0.825 | 1.086 | 1.254 | 1.342 | 1.459 | 1.552 |

## J.2 Class-conditional image experiments

We supplement Table 1 and Figure 7 with additional results and samples.

We plot all four metrics – FID, CMMD, average log rewards and average diversity – across different number of function evaluations (NFEs) for the three settings considered in Section 5.1. The plots show that across the three settings for most values of NFEs, DTS matches the target distribution more accurately compared to other methods (lowest FID and CMMD).

We also present random samples for each method and setting in Figures 13 to 14. We observe the same trend as noticed in Figure 7 – gradient-based guidance like DPS can be unstable leading to unnatural images, while SMC-based methods show signs of mode collapse with low average diversity and high average rewards. DTS balances both diversity and high rewards effectively by closely matching the true posterior distribution.

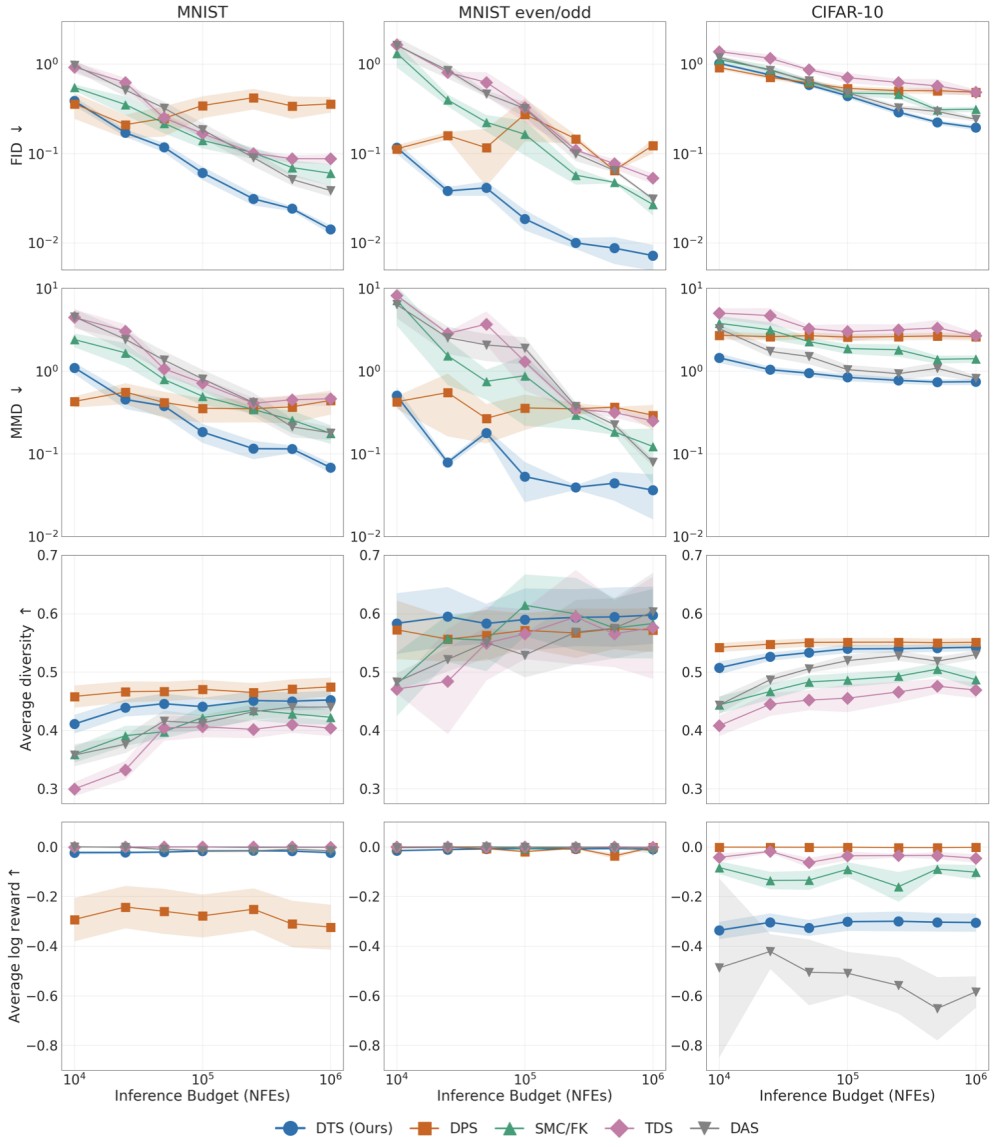

Figure 12: Distribution level metrics vs. number of function evaluations for different methods on MNIST single digit generation averaged over all 10 digits (left), MNIST odd and even digit generation (center), and CIFAR-10 single class generation averaged over all 10 digits (right). All methods were evaluated with 5000 generated samples per class. Metrics reported: FID (lower is better), CMMD (lower is better), Average log rewards (higher is better), and average diversity (higher is better).

Figure 13: MNIST posterior samples generated using different methods for digits 0-9, even and odd.

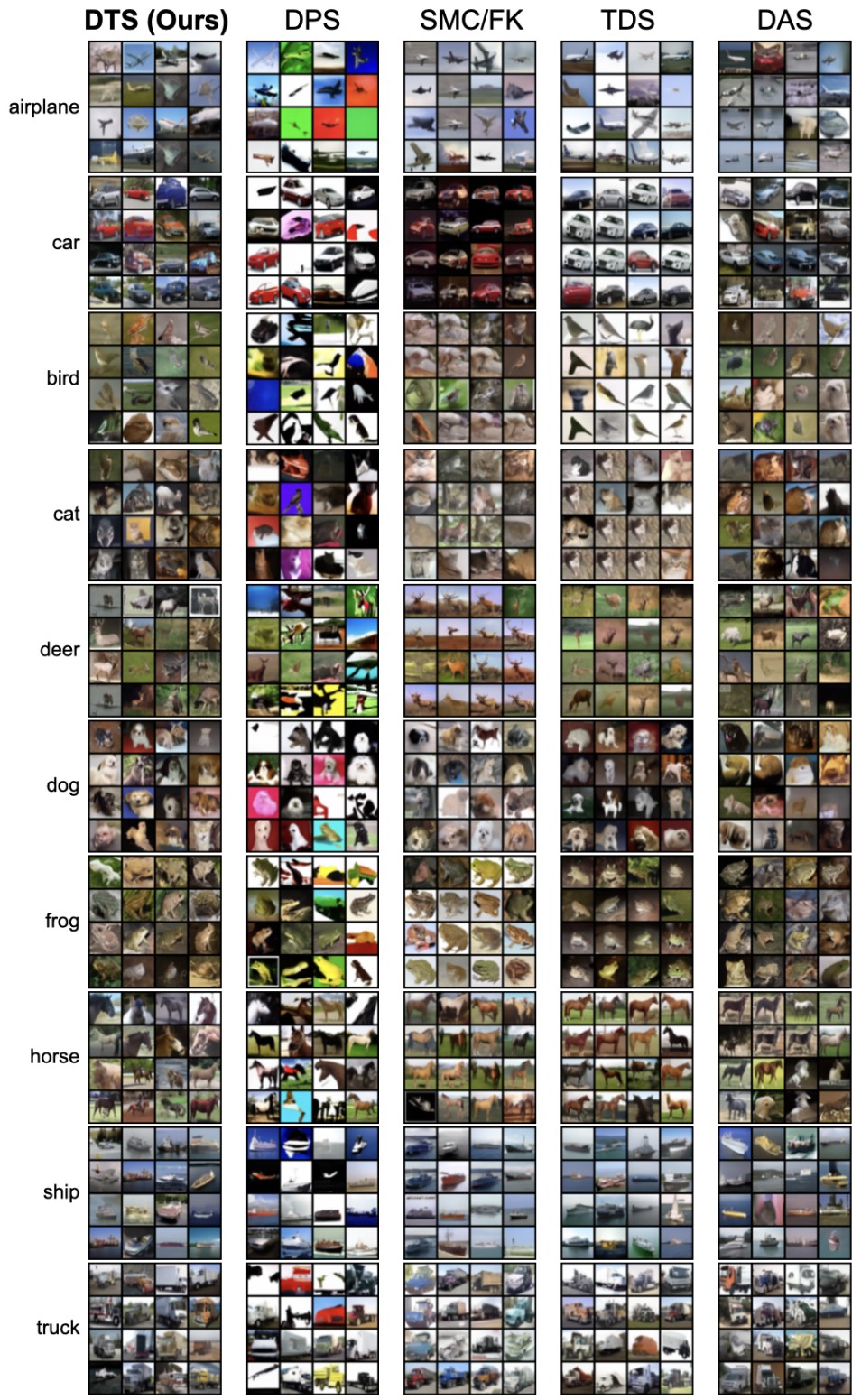

Figure 14: CIFAR-10 posterior samples generated using different methods for all classes.

## J.3 Text-to-image examples

We present more samples for qualitative analysis. Figure 15 shows how samples change with increasing amount of inference-time compute, providing visual evidence for the quantitative results from Figure 8. Figures 16 to 18 shows text-image pairs testing different concepts such as artistic style, spatial arrangement and object count.

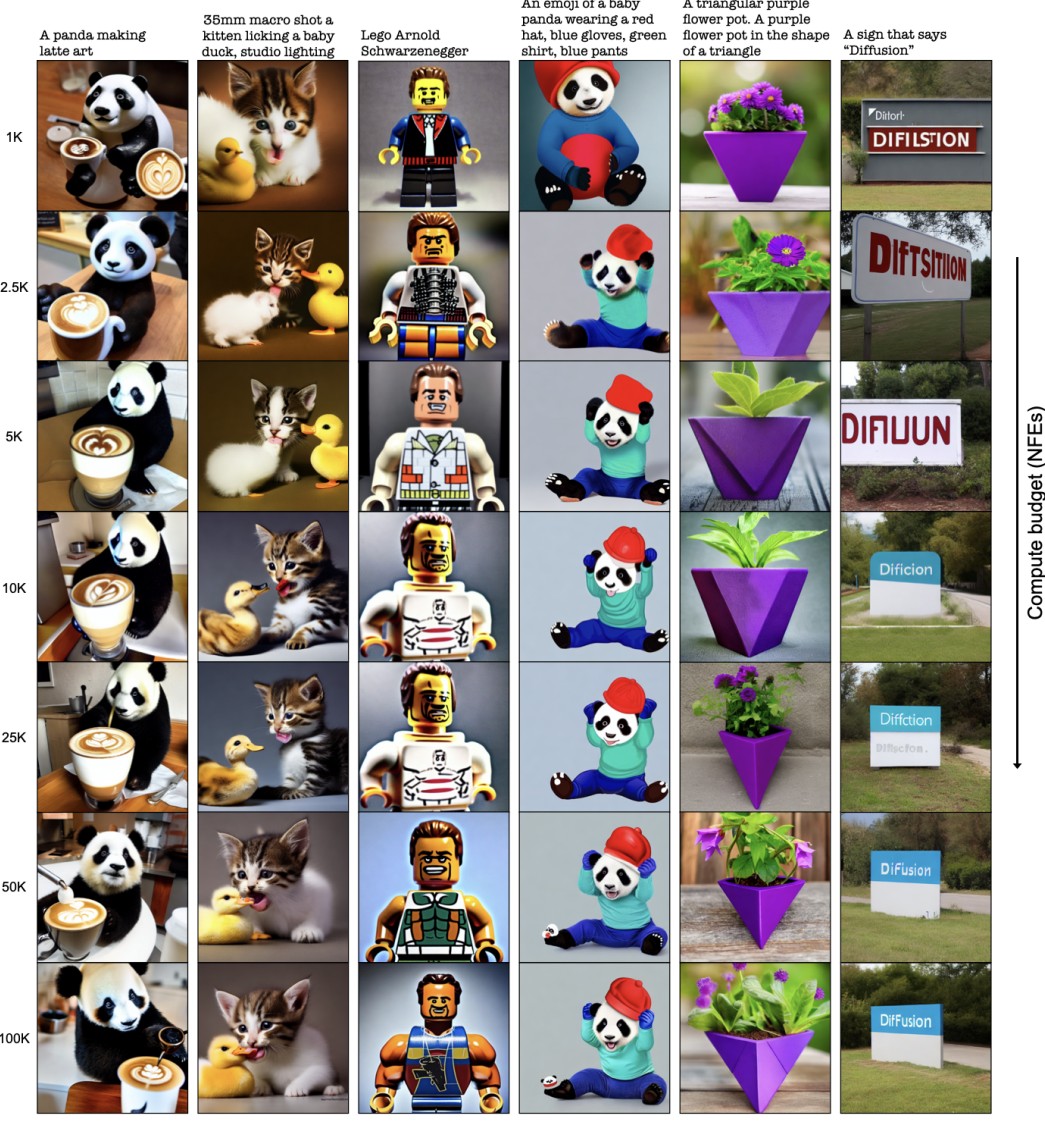

Figure 15: Text-image pairs from Figure 1 with increasing amount of inference-time compute, measured in number of function evaluations (NFEs) of the diffusion model.

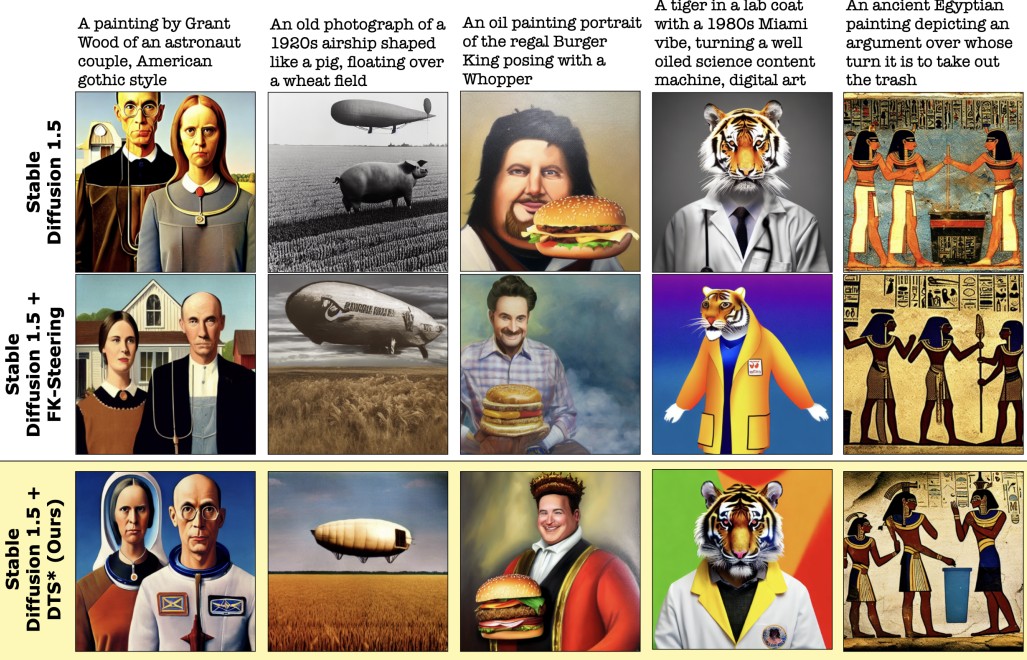

Figure 16: Sample text-image pairs using Stable Diffusion v1.5 and ImageReward as the guiding function for prompts requiring a specific artistic style. Samples are picked at random for each method and prompt.

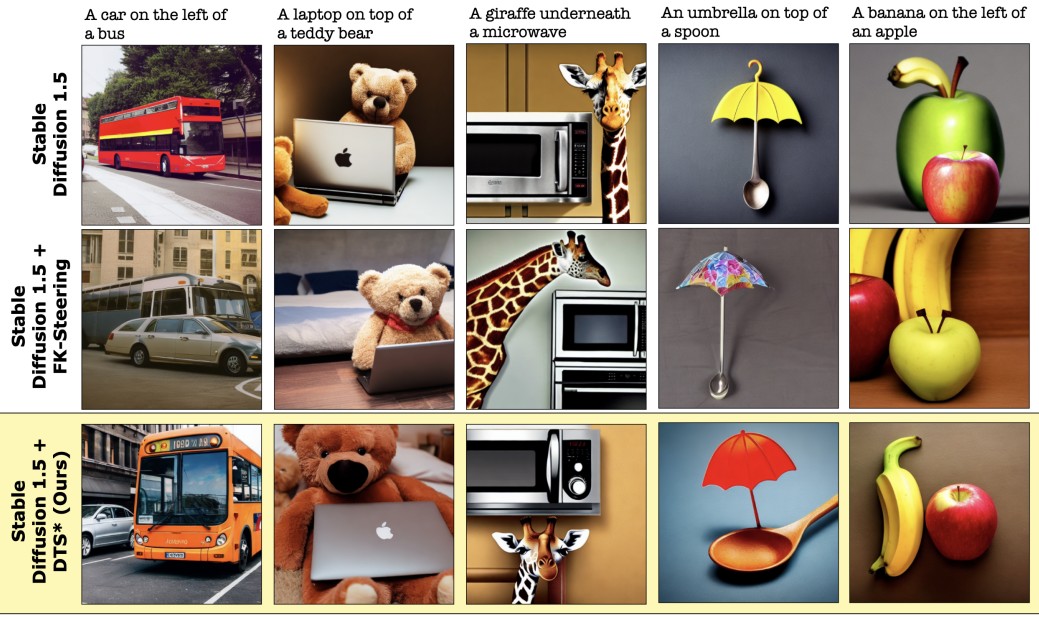

Figure 17: Sample text-image pairs using Stable Diffusion v1.5 and ImageReward as the guiding function for prompts requiring specific spatial relationships between objects. Samples are picked at random for each method and prompt.

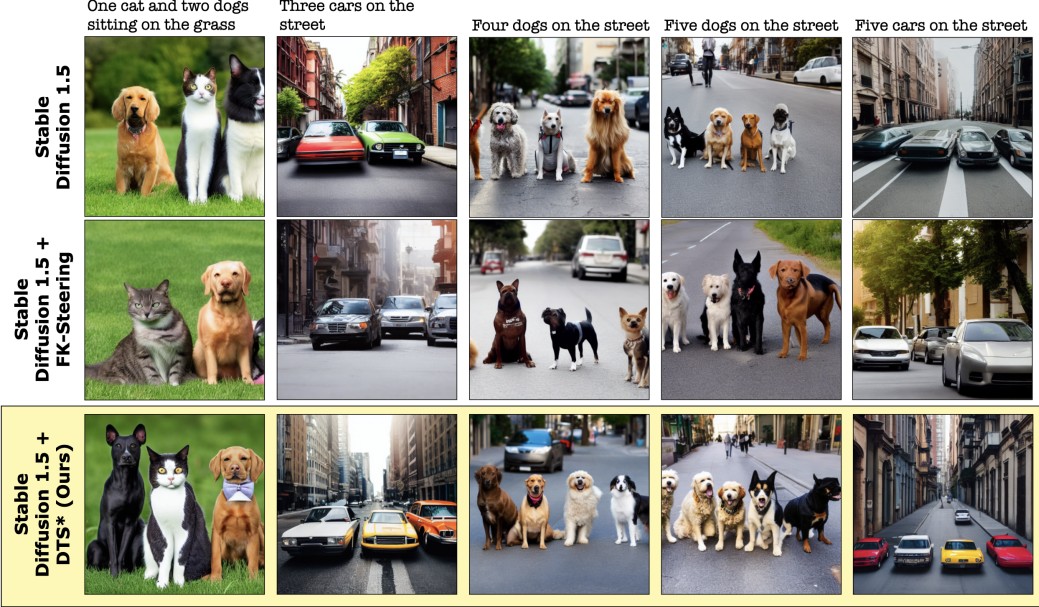

Figure 18: Sample text-image pairs using Stable Diffusion v1.5 and ImageReward as the guiding function for prompts requiring specific object counts. Samples are picked at random for each method and prompt.

## J.4 Text completion examples

We present additional text completions for the base MDLM model, FK-Steering and DTS$^{\star}$ in Figure 19. We also evaluate the text samples using ChatGPT-4o, since reward in this case does not necessarily capture quality, and previous works [47] suggest that LLM evaluation aligns better with human judgement.

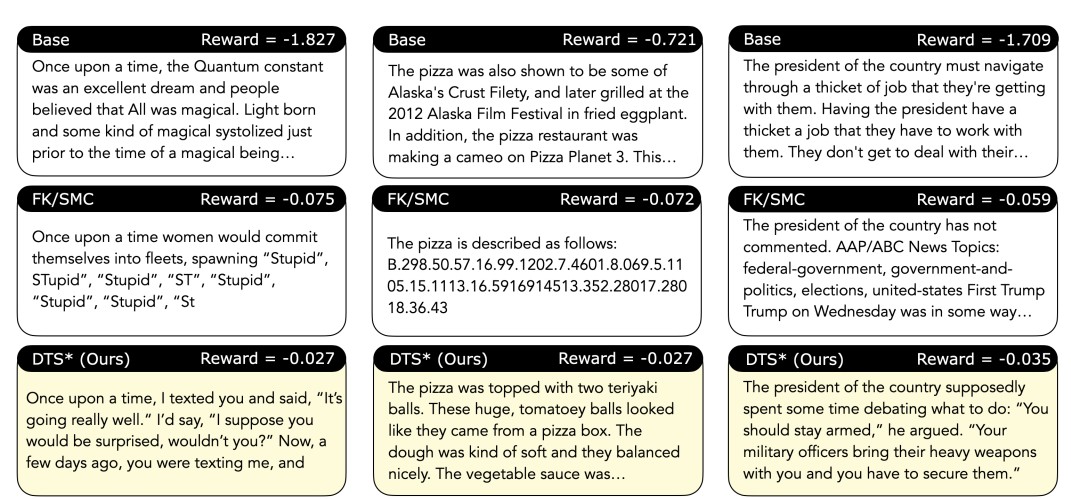

Figure 19: Sample text completions using MDLM and a CoLA classifier as reward. Samples are picked at random for each method and prompt.

Table 4: Text samples generated for $2^{17}$ NFEs with the CoLA reward (Section 5.3) are scored by ChatGPT-4o between $0$ and $1$ following a rubric. Results are reported as mean$\pm$std by averaging over 3 completions for each of the 15 prompts. DTS$^\star$ improves $13.7\%$ over the best baseline.

| Algorithm $\rightarrow$ | Best-of-N | FK/SMC | DTS$^\star$ (Ours) |
|---|---|---|---|
| ChatGPT-4o score | $0.283 \pm 0.020$ | $0.216 \pm 0.019$ | $0.322 \pm 0.022$ |

---

**Prompt for LLM evaluation**

You are an expert language evaluator. Your task is to assess the overall quality of a text sample based on the following criteria:
1. **Fluency**: Is the text generally grammatically correct and understandable?
2. **Diversity**: Does the text show some variety in vocabulary or sentence structure, even if modest?
3. **Plausibility**: Is the content mostly coherent, logical, and believable in its context?
Please be constructive and fair in your evaluation — highlight the strengths of the sample as well as areas that could be improved. Some minor errors or simplicity are acceptable if the overall message is clear.
For each sample, provide a brief explanation touching on each criterion, followed by a quality score between **0.0** (very poor) and **1.0** (excellent). Format the final score using \boxed{score}.
—
**Example 1 (High Quality)**
Sample: """The shimmering lake reflected the golden hues of the setting sun as birds glided across the surface, their wings slicing through the calm evening air."""
Evaluation: - **Fluency**: The sentence flows smoothly with proper grammar and elegant phrasing. - **Diversity**: The vocabulary is rich ("shimmering", "glided", "golden hues"), and sentence structure is varied. - **Plausibility**: The description is vivid and realistic. \boxed{0.95}
—
**Example 2 (Low Quality but Understandable)**
Sample: """Lake is shiny. Birds is fly. The sky is color like orange. It is lake."""
Evaluation: - **Fluency**: Some grammatical issues ("birds is fly", "it is lake"), but basic meaning is understandable. - **Diversity**: Limited vocabulary and repetition. - **Plausibility**: The message is simple but not nonsensical. \boxed{0.25}
—
Now evaluate the following sample:
Sample: """{sample}"""
Evaluation:

