# OpenReview forum: "Diffusion Tree Sampling: Scalable inference‑time alignment of diffusion models"
_NeurIPS.cc/2025/Conference — NeurIPS 2025 poster_

### Official Review · Reviewer_t7aZ · 2025-06-25

**Clarity:** 3
**Significance:** 3
**Originality:** 3
**Rating:** 5
**Confidence:** 4

**Summary:**

This paper introduces Diffusion Tree Sampling (DTS), a novel method for inference-time alignment of diffusion models that is inspired by Monte Carlo Tree Search (MCTS). The authors frame the iterative denoising process as a search problem on a tree, where nodes represent noisy states. By performing rollouts to the terminal state and backing up the observed rewards through the tree, DTS iteratively refines the value estimates of intermediate states. This approach addresses a key weakness of prior methods: the reliance on inaccurate, one-step value estimations, especially at high noise levels. DTS is designed as an "anytime" algorithm that reuses past computations to progressively improve sample quality. A greedy search variant, DTS*, is also proposed for optimization tasks.

**Questions:**

- The key concern on this method is clearly its scalability with memory. Could the authors provide an analysis comparing the memory footprint of DTS to baselines like SMC for a fixed task?
- In the ordinary DTS without UCT exploration, is the stochasticity from the Boltzmann sampling sufficient to ensure the tree's exploration, if the modes of the reward is quite diverse?
- I would suggest including a brief discussion on other related methods on this topic, e.g. SGS [1,2], and recent advances in SMC [3,4], etc..


### Minor Issues
- It seems in the proofs in Appendix D.1, several $V$ lacks their corresponding subscripts, making it hard to follow.
- On line 178, the root should be changed from $x_0$ to $x_t$.

[1] Wu, Zihui, et al. "Principled probabilistic imaging using diffusion models as plug-and-play priors." Advances in Neural Information Processing Systems 37 (2024): 118389-118427.\
[2] Coeurdoux, Florentin, Nicolas Dobigeon, and Pierre Chainais. "Plug-and-play split Gibbs sampler: embedding deep generative priors in Bayesian inference." IEEE Transactions on Image Processing (2024).\
[3] Skreta, Marta, et al. "Feynman-kac correctors in diffusion: Annealing, guidance, and product of experts." arXiv preprint arXiv:2503.02819 (2025).\
[4] Chen, Haoxuan, et al. "Solving inverse problems via diffusion-based priors: An approximation-free ensemble sampling approach." arXiv preprint arXiv:2506.03979 (2025).\
[5] He, Jiajun, et al. "RNE: a plug-and-play framework for diffusion density estimation and inference-time control." arXiv preprint arXiv:2506.05668 (2025).

**Ethical Concerns:**

["NO or VERY MINOR ethics concerns only"]

**Final Justification:**

I am updating the score to 5 (Accept) upon their promised revision, in light of the overall quality and novel idea of the paper.

**Limitations:**

Limitations are adequately addressed.

**Paper Formatting Concerns:**

The paper seems conforming to the formatting instructions.

**Quality:**

3

**Strengths And Weaknesses:**

### Strengths
- The concept of applying MCTS-like method to diffusion model alignment is novel and provides a new paradigm for guidance. One key feature of the algorithm is "anytime", that is reusing information from all past simulations to improve value estimates.
- The paper provides comprehensive empirical evidence of DTS's effectiveness. It is shown to match the performance of leading baselines on a variety of tasks, including image generation, text-to-image, etc.

### Weaknesses
- The method requires a large amount of memory, including the entire search tree and intermediate states. This could lead to prohibitive computational cost for large-scale tasks.
- It would be beneficial to include more details on DTS*, the proposed variant of DTS, discussing on how the modified algorithm fosters exploration.

---

> ### Author Rebuttal · Authors · 2025-07-31
>
> We thank the reviewer for their careful review and positive assessment of our work. We address your concerns below, and also draw attention to our response to other reviewers for additional experiments - text diffusion with LLM evaluation (reviewer 1bSF), image inpainting (reviewer ULya), and comparison between DTS and DTS* (reviewer T4KC). Note that this year, NeurIPS does not allow posting any links or documents in the rebuttal.
>
> > Comparing memory footprint and wall clock runtime
>
> Below, we provide peak memory usage for each method in the text-to-image generation experiment using Stable Diffusion 1.5 and ImageReward. We perform this experiment with 40 different prompts and average the results. The numbers are reported for a single NVIDIA A100 GPU with 40GB of memory using the pytorch profiler.
>
> **Peak Memory Usage (MB):**
>
> We observe that the memory usage of DTS* is much lower compared to FK/SMC, and is more comparable to best-of-N. Initially, DTS* uses slightly more memory than best-of-N, but with more compute, it tilts in favor of DTS*, showing that it actually scales better in terms of memory.
>
> The main memory usage of DTS* is the tree as noted by the reviewer, which stores 64x64x4 latents. Note that currently the entire tree is kept on the GPU, and we can save GPU memory by offloading the tree to the CPU. For best-of-N, the memory usage peaks at the end when all N candidates need to be decoded using the VAE into 512x512x3 images and evaluated using the reward function to pick the best sample. This problem does not occur with DTS*, since the rewards are calculated when traversal reaches the terminal state, and multiple high-resolution images do not need to be kept in memory. For FK/SMC, this cost is very high since all N particles need to be decoded into 512x512x3 images, then evaluated using the reward function *at each resampling step*. In general, for images and text, the reward function itself is a large pretrained model, and computing the reward requires a non-trivial amount of compute.
>
> *Peak memory usage (MB), lower is better*
>
> | **Method / NFEs** | 1000 | 2500 | 5000 | 10000 | 25000 | 50000 | 100000 |
> |:----------|-----:|-----:|-----:|------:|------:|------:| ------:|
> | **DTS\* (Ours)** | 8059 | 9140 | 9460 | 10103 | **12025** | **15226** |  **17318** |
> | **Best-of-N** | **8056** | **8331** | **8789** | **9705** | 12453 | 17033 | 19782 |
> | **FK/SMC** | 27167 | 27348 | 27559 | 27974 | 29220 | 31296 | 33948 |
>
> **Wall clock time (seconds):**
>
> In terms of wall clock time, DTS* is approximately 2.5x slower than the other methods for the same number of function evaluations (NFEs), since it is harder to parallelize. Note that the runtime reflects tree building from scratch - once the tree is built, repeated sampling is just pointer chasing and is near instantaneous (whereas for other methods repeated sampling would require starting the entire procedure again).
>
> We discuss two important considerations regarding the runtime:
>
> 1) When taking into account their performance, DTS* is still preferable in terms of wall clock time since it outperforms the baselines and needs much fewer NFEs to reach the same level of performance. As noted in the paper and the table below, for the text-to-image task, it needs roughly 5x fewer NFEs to match the reward of the best performing baseline (best-of-N) at 100k NFEs. In other words, to match the performance of best-of-N at 100k NFEs, DTS* is approximately 2x faster in terms of wall clock time.
>
> 2) We can significantly reduce the runtime of DTS/DTS* by using multiple actors in parallel that traverse and update a shared tree (similar to AlphaZero training). This will give a roughly linear speedup in runtime at the cost of increased memory for copying the diffusion model (the tree is not copied since it can be shared among all actors, hence increase in memory will be sublinear in terms of the number of actors). This represents an inherent trade-off between memory usage and runtime. We did not implement a distributed pipeline since it requires some engineering effort, but it is a viable option.
>
> *Wall clock time (seconds), lower is better*
>
> | **Method / NFEs** | 1000 | 2500 | 5000 | 10000 | 25000 | 50000 | 100000 |
> |:----------|-----:|-----:|-----:|------:|------:|------:| ------:|
> | **DTS\* (Ours)** | 52 | 132 | 267 | 543 | 1372 | 2763 | 5513 |
> | **Best-of-N** | 21 | 50 | 96 | 188 | 461 | 926 | 1910 |
> | **FK/SMC** | 23 | 55 | 107 | 213 | 535 | 1073 | 2204 |
>
> *ImageReward score, higher is better*
>
> | Method | 1000 | 2500 | 5000 | 10000 | 25000 | 50000 | 100000 |
> |:----------|---------:|---------:|---------:|---------:|---------:|---------:| ---------:|
> | **DTS\* (Ours)** | 1.148 | 1.309 | 1.407 | 1.506 | 1.605 | 1.665 | 1.704 |
> | **Best-of-N** | 1.140 | 1.269 | 1.350 | 1.417 | 1.478 | 1.530 | 1.564 |
> | **FK/SMC** | 1.182 | 1.221 | 1.324 | 1.338 | 1.423 | 1.506 | 1.528 |
>
> > It would be beneficial to include more details on DTS*, the proposed variant of DTS, discussing on how the modified algorithm fosters exploration.
>
> DTS* involves changing the selection rule, where instead of sampling children proportional to exponentiated values, we pick the child with the highest value at each node. We discuss some considerations in Section 4.3.  Given the reviewer’s suggestion (and also those of reviewer ULya), we will elaborate on some of the details a bit more with the extra page allowed in the final version of the paper. DTS* inherently does not foster exploration, which is why we use UCT to add some exploration in the tree traversal.
>
> > In the ordinary DTS without UCT exploration, is the stochasticity from the Boltzmann sampling sufficient to ensure the tree's exploration, if the modes of the reward is quite diverse?
>
> The reviewer raises an excellent point. In the sampling case, since DTS selects children using Boltzmann sampling, it inherently has exploration ability since there is a non-trivial probability of sampling suboptimal children. Empirically, we found no performance difference when using UCT with DTS over a range of exploration coefficients. This is also supported theoretically by recent work in the bandit setting, where if the goal is to sample from the Boltzmann distribution of the reward, simply picking arms based on the Boltzmann distribution of the estimated reward without explicit exploration leads to optimal regret bounds [1].
>
> > I would suggest including a brief discussion on other related methods on this topic, e.g. SGS [A,B], and recent advances in SMC [C,D], etc..
>
> We thank the reviewer for pointing out these works. We will include the following discussion in the related works section.
>
> “Recent plug-and-play samplers [A, B] also use Tweedie’s formula to get biased values and use compute-heavy MCMC/Gibbs chains for inverse problems. Skreta et al. [C] use SMC correctors to re-weight particles each step which inherits the same problems of diversity collapse and low effective sample size when the weights have high variance. Chen et al. [D] propagate a diffusion-PDE ensemble, which is accurate but can be compute-intensive; while He et al. [E] steer paths via high-variance density-ratio estimates. Our principled search method produces high quality samples by using unbiased values and works with arbitrary rewards.”
>
> > It seems in the proofs in Appendix D.1, several V lacks their corresponding subscripts, making it hard to follow.
>
> > On line 178, the root should be changed from x_0 to x_T
>
> Thank you for pointing out these typing errors. The root should indeed be $x_{T+1}$ in line 178, and we will add subscripts to the value function in the appendix.
>
> *[1] Pedramfar, Mohammad, and Siamak Ravanbakhsh. "Multi-Armed Sampling Problem and the End of Exploration." arXiv preprint arXiv:2507.10797 (2025).*
>
> *[A] Wu, Zihui, Yu Sun, Yifan Chen, Bingliang Zhang, Yisong Yue, and Katherine Bouman. "Principled probabilistic imaging using diffusion models as plug-and-play priors." Advances in Neural Information Processing Systems 37 (2024): 118389-118427.*
>
> *[B] Coeurdoux, Florentin, Nicolas Dobigeon, and Pierre Chainais. "Plug-and-play split gibbs sampler: embedding deep generative priors in bayesian inference." IEEE Transactions on Image Processing 33 (2024): 3496-3507.*
>
> *[C] Skreta, Marta, Tara Akhound-Sadegh, Viktor Ohanesian, Roberto Bondesan, Alan Aspuru-Guzik, Arnaud Doucet, Rob Brekelmans, Alexander Tong, and Kirill Neklyudov. "Feynman-Kac Correctors in Diffusion: Annealing, Guidance, and Product of Experts." In Forty-second International Conference on Machine Learning, 2025.*
>
> *[D] Chen, Haoxuan, Yinuo Ren, Martin Renqiang Min, Lexing Ying, and Zachary Izzo. "Solving inverse problems via diffusion-based priors: An approximation-free ensemble sampling approach." arXiv preprint arXiv:2506.03979 (2025).*
>
> *[E] He, Jiajun, José Miguel Hernández-Lobato, Yuanqi Du, and Francisco Vargas. "RNE: a plug-and-play framework for diffusion density estimation and inference-time control." arXiv preprint arXiv:2506.05668 (2025).*

---

> > ### Comment · Reviewer_t7aZ · 2025-08-05
> >
> > I am updating the score to 5 (Accept) upon their promised revision, in light of the overall quality, novel idea of the paper, and their detailed rebuttal.

---

> > > ### Author Response · Authors · 2025-08-08
> > >
> > > We thank the reviewer for their feedback and for helping us improve our paper. The memory and runtime considerations are certainly an important consideration - we will include these results in the final version of our paper.

---

> ### Author Response · Authors · 2025-08-04
>
> Dear reviewer,
> We appreciate your time and feedback on our work. As the discussion period comes to an end, we wanted to confirm if our rebuttal addressed your concerns. We showed that DTS scales more favorably in terms of memory compared to other methods, and achieves the same performance as the best baseline with less wall clock time. We also answered all of the questions regarding exploration, DTS*, and some related works.
>
> If there are any outstanding questions, we will be happy to answer them!

---

### Official Review · Reviewer_T4KC · 2025-06-25

**Clarity:** 2
**Significance:** 3
**Originality:** 3
**Rating:** 4
**Confidence:** 4

**Summary:**

The paper presents a Monte Carlo Tree Search algorithm for inference-time alignment of diffusion models. Authors cast inference-time alignment as a search problem that reuses past computations, and intoroduce an algorithm that samples from reward-aligned target density by propagating rewards through the search tree. Authors present a theoretical motivation for their method and experimentally show that the method achieves better quality of the samples than considered baselines, as well as better scales with compute.

**Questions:**

0) See Weaknesses.

1) Line 270: does $10^6$ denote the number of function evaluations used to generate each image, or the whole set of samples? If it is the NFE for a single image, this seems unreasonably high for such simple datasets. Did the authors measure the GPU time needed to generate one image? The same question for the NFEs denoted in plots in Figures 6, 7, 8.

2) Algorithm 1 describes how the search tree is built. Are final samples used for evaluation the same as the leaves created during the construction of the tree, or are they obtained by resampling paths inside the tree after construction? I did not find the description of how the final samples for evaluation are obtained in the paper, I think it would be better to clarify explicitly.

3) At first I struggled to understand the motivation and message of Section 3.1 in the text, as it feels a bit disconnected from the rest of the work. I think that explicitly explaining in the text how the value estimates by themselves can help with the problem at hand (e.g. how they can be used for guidance) could improve the readability of this section.

**Ethical Concerns:**

["NO or VERY MINOR ethics concerns only"]

**Final Justification:**

Despite the weaknesses in presentation I outlined in my review, I believe that this is a solid work. I'm glad that my comments and suggestions about the clarity of theoretical results and comparisons between DTS and DTS* were helpful to the authors for the purpose of improving the paper. I decided to maintain my score and remain positive towards accepting this work.

**Limitations:**

No concern.

**Paper Formatting Concerns:**

No concern.

**Quality:**

2

**Strengths And Weaknesses:**

**Strengths**:

The paper is generally well-structured and easy to read (except for some theoretical parts and evaluation details, see weaknesses and questions below). The idea of applying MCTS for inference-time alignment of diffusion models is novel and, in my opinion, very interesting and promising. The experimental evaluation and results are solid. I also did enjoy the illustrative synthetic experiments in Section 4.4, although I would suggest moving Figure 9 from Appendix H.1 to the main text as it would better help illustrate the performance of the proposed method (in Figure 4 the difference with some baselines is visually less obvious).

**Weaknesses**:

My main concern for this paper is the presentation of theoretical background and analysis for the proposed method.

First of all, up to my knowledge, the definition of soft value in Section 2 is not the most common one used in KL-regularized RL problems. Usually, the soft optimal value is defined through the expected reward minus KL term, where the expectation is taken with respect to the optimal policy, not the reference policy (which is $p_\theta$ in this paper). In the context of RL-based diffusion fine-tuning see, e.g., Section 2.2 of [1]. Although, from my understanding, the aforementioned definitions should produce equivalent functions, authors should either reference some previous literature that studies the exact same definition, or elaborate and provide more context for the reader. Authors reference two papers on entropy-regularized RL, which is a related setting, but not exactly the same, and study a different problem.

Secondly, theoretical derivations in Appendix D seem valid to me, however, whether the Algorithm 1 in Appendix E directly follows them is quite unobvious. E.g., optimal policy in Equation 6 is proportional to $p_\theta$ times the exponent of value, while in line 9 of Algorithm 1 the next state is sampled proportionally to just the exponent of value. Similarly, the mentioned soft Bellman equation in Equation 5 expresses the value through the expectation of exponentiated values in the next states with respect to the reference policy, while soft backup in Algorithm 1 (line 26) just takes the sum of exponentiated values in the next states, not the expectation with respect to $p_\theta$. Do I understand correctly that this is due to the fact that the expectation with respect to $p_\theta$ is thought to be estimated via Monte Carlo, since paths in the tree are sampled using $p_\theta$? If this is the case, the authors should definitely elaborate on this more in their theoretical derivations. If not, then I do not really understand the theory behind the proposed algorithm. Also, if Monte Carlo estimate is used in line 26, shouldn't there be the logarithm of the mean, not the logarithm of the sum?

Addressing these points will help to substantially improve the theoretical part of the work and its clarity in my opinion.

The presented connection to GFlowNets in Appendix C is interesting and insightful, however, there is a caveat in how it is written in the text. If it is the usual GFlowNet task of sampling from the target distribution, and $F$ denotes the log-flow, then the soft Bellman equation in Appendix C should use the backward policy $P_B(s \mid s')$, not the trainable forward policy $P_\theta(s' \mid s)$. See [2] for the analysis of connections to entropy-regularized RL and how soft Bellman equations are expressed through GFlowNet flows and policies. However, [3] puts the task of fine-tuning of diffusion models in the GFlowNet framework, and in such case the reference policy will replace the backward policy in equations. In this case the approach proposed in this paper will indeed be very related to the one presented in [4], which is another insightful observation and connection made by the authors.

I also have some minor points related to the Experiments section. I think that putting some direct comparison of DTS and DTS* would help the reader to better understand the benefits of each. It looks a bit odd that some experiments in the paper present the results only for DTS, and others only for DTS*. In addition, it would be good to put some references to previous literature near the statement "reward functions in text domains are particularly susceptible to over-optimization, often resulting in repetitive outputs", as making such observation from just one experiment in the paper is not well enough validated.

References:

[1] Uehara et al. Understanding Reinforcement Learning-Based Fine-Tuning of Diffusion Models: A Tutorial and Review. 2024\
[2] Tiapkin et al. Generative Flow Networks as Entropy-Regularized RL. AISTATS 2024\
[3] Venkatraman et al. Amortizing intractable inference in diffusion models for vision, language, and control. NeurIPS 2024\
[4] Morozov et al. Improving GFlowNets with Monte Carlo Tree Search. ICML 2024 SPIGM Workshop

---

> ### Author Rebuttal · Authors · 2025-07-31
>
> We thank the reviewer for their detailed review and positive assessment of our work. We address your concerns below, and also draw attention to our response to other reviewers for additional experiments - text diffusion with LLM evaluation (reviewer 1bSF), image inpainting (reviewer ULya) and comparison of memory and runtime (reviewer t7aZ). Note that this year, NeurIPS does not allow posting any links or documents in the rebuttal.
>
> > Definition of soft value function.
>
> The understanding of the reviewer is correct, these two definitions are equivalent. Indeed, we use the same definition of the value function as in Uehara et al. [1] (Equation 6 in Section 2.1). We will add a reference to this work in Section 2 as suggested by the reviewer.
>
> > Sampling and expected value in backup
>
> The reviewer understands correctly. Since the nodes are expanded using $p_\theta$, when we perform selection based on the exponentiated values, we are effectively sampling from the product $p_\theta \exp(V_t(x_t))$. The same reasoning is applied in the backup, since the Monte Carlo estimate uses samples drawn from $p_\theta$. We mention that children are drawn from $p_\theta$ in step 1 of the proof, but we agree with the reviewer that it should be made more explicit, especially when explaining the backup operation. We will update this in the paper.
>
> Thank you for pointing out the typing error in Algorithm 1, line 26 - instead of sum, it should be mean over the children (dividing by the number of children $|C_t|$).
>
> > Connection to GFlowNets
>
> Thank you for pointing this out. The soft Bellman equation in Appendix C should indeed use the backward policy. In the case of diffusion, $P_B(s | s’)$ is the noise process that moves from data to noise space. We think the reviewer may find Figure 2 from He et al. [2] helpful. We will correct this in the paper.
>
> > Some experiments in the paper present the results only for DTS, and others only for DTS*
>
> There are two different types of experiments considered in our paper: (1) posterior sampling experiments (the illustrative 2D and class-conditional image generation experiments), and (2) what one may call “high reward search” experiments (text-to-image and text diffusion experiments). The nature of these two distinct tasks and prior literature motivate the choice of algorithms. DTS is designed to tackle problem (1) and DTS* for problem (2).
>
> In the first setting, there is a well-defined and tractable ground truth distribution (the reward-tilted distribution in the 2D case and the class-conditioned image distribution in the image case). DTS and the baselines considered in these experiments are designed to sample approximately from this target distribution; hence, we report distribution-based metrics.
>
> In the second setting, the goal is to produce the single best sample that achieves both high likelihood under the pretrained model and high reward [3,4]. To generate an image for a given caption, for example, we care about having a single good image. DTS and the other baselines from the previous setting are not designed to produce a single good sample, unlike DTS*. Therefore, we only consider each of these methods in its respective setting. We will explain the difference between the two settings more clearly in the paper.
>
> Following the reviewer’s request, we compare DTS and DTS* on the text-to-image setting using Stable Diffusion 1.5 with ImageReward. Due to time limitations, we limit this experiment to 40 prompts from DrawBench. We picked complex prompts where the base model struggles to generate high reward images to demonstrate the difference more clearly. For DTS, we sample 16 images from the tree and pick the one with the highest reward. The results show that there is a noticeable gap, since DTS spends more compute expanding suboptimal paths whereas DTS* aggressively searches only the highest reward regions (except for the small exploration bonus).
>
> *ImageReward, higher is better*
>
> | **Method / NFEs**   | **1000** | **2500** | **5000** | **10000** | **25000** | **50000** |
> |:---------|----------:|----------:|----------:|----------:|----------:|----------:|
> | **DTS**      | 0.774 | 1.025 | 1.064 | 1.112 | 1.214 | 1.238 |
> | **DTS\***   | **0.825**  | **1.086**  | **1.254**   | **1.342**  | **1.459**  | **1.552**  |
>
>
> > References for supporting the statement about text reward functions
>
> Following the reviewer’s suggestion, we will add the following references to support the statement:
>
> Holtzman et al. [5] highlight that text generation strategies that optimize for perplexity of some learned model often lead to dull, repetitive, and unnatural text. Hashimoto et al. [6] observe that metrics often fail to capture the trade-offs that human evaluators make, such as quality vs. diversity. The excellent survey by Celikyilmaz et al. [7] discusses that statistical evaluation of text generally correlates poorly with human judgments and does not necessarily follow a linear scale. We add evaluations of the text samples using ChatGPT 4o in our response to reviewer 1bSF, following previous work which shows these evaluations tend to have higher correlation with human judgement [8].
>
> > Meaning of NFEs
>
> The NFE values in all our experiments refer to the total number of function evaluations used in the algorithm. We test all methods for different values of NFEs to study how these methods scale with more compute. For a concrete example, if we are comparing all methods at 10^6 NFEs with 50 denoising steps per sample, best-of-N and the SMC-based methods would use 20000 (=10^6 / 50) particles, and DTS/DTS* would have 10^6 total nodes in the tree. Each individual sample is still obtained using the same number of denoising steps (50 in this example). We refer to our response to reviewer t7aZ for comparison of runtime and memory usage of different methods.
>
> > How are the final samples obtained?
>
> Once the tree is constructed, we perform a final selection process (depending on DTS/DTS*) by traversing the tree from the root to obtain the final samples. These would be a subset of the leaves constructed in the tree. Following the reviewer’s suggestion, we will add a separate Algorithm 2 that, given a constructed tree, performs selection to obtain the final samples.
>
> > Clarifying motivation and message of Section 3.1
>
> Thank you for the feedback. We will add some motivation for studying value functions to connect it with the rest of the paper and improve clarity. We will add the text below to the beginning of Section 3.1:
>
> “The central problem for inference-time alignment of diffusion models is accurate credit assignment during denoising to evaluate which noisy states have the potential to lead to higher reward samples. This notion is captured by value functions as defined in Equation 5. The discussion above explains how most existing methods use the value function in some form, however, since they are generally intractable to compute, certain approximations are commonly used in literature.”
>
> We will add the following text to the end of Section 3.1 to connect the results of Figure 2 with the rest of the paper:
>
> “These inaccurate value estimates lead to biased guidance during denoising, resulting in samples that may not approximate the target distribution accurately.”
>
> > Moving Figure 9 from Appendix H.1 to the main text
>
> We had to move Figure 9 to the appendix due to space constraints. The final version of the paper allows one additional page, and if accepted, we plan to move Figure 9 to the main paper.
>
>
> *[1] ​​Uehara, Masatoshi, Yulai Zhao, Chenyu Wang, Xiner Li, Aviv Regev, Sergey Levine, and Tommaso Biancalani. "Inference-time alignment in diffusion models with reward-guided generation: Tutorial and review." arXiv preprint arXiv:2501.09685 (2025).*
>
> *[2] He, Haoran, Emmanuel Bengio, Qingpeng Cai, and Ling Pan. "Random Policy Evaluation Uncovers Policies of Generative Flow Networks." In Forty-second International Conference on Machine Learning, 2025.*
>
> *[3] Singhal, Raghav, Zachary Horvitz, Ryan Teehan, Mengye Ren, Zhou Yu, Kathleen McKeown, and Rajesh Ranganath. "A General Framework for Inference-time Scaling and Steering of Diffusion Models." In Forty-second International Conference on Machine Learning, 2025.*
>
> *[4] Ma, Nanye, Shangyuan Tong, Haolin Jia, Hexiang Hu, Yu-Chuan Su, Mingda Zhang, Xuan Yang et al. "Inference-time scaling for diffusion models beyond scaling denoising steps." arXiv preprint arXiv:2501.09732 (2025).*
>
> *[5] Holtzman, Ari, Jan Buys, Li Du, Maxwell Forbes, and Yejin Choi. "The Curious Case of Neural Text Degeneration." In International Conference on Learning Representations, 2020.*
>
> *[6] Hashimoto, Tatsunori B., Hugh Zhang, and Percy Liang. "Unifying Human and Statistical Evaluation for Natural Language Generation." In Proceedings of the 2019 Conference of the North American Chapter of the Association for Computational Linguistics: Human Language Technologies, Volume 1 (Long and Short Papers), pp. 1689-1701. 2019.*
>
> *[7] Celikyilmaz, Asli, Elizabeth Clark, and Jianfeng Gao. "Evaluation of text generation: A survey." arXiv preprint arXiv:2006.14799 (2020).*
>
> *[8] Liu, Yang, Dan Iter, Yichong Xu, Shuohang Wang, Ruochen Xu, and Chenguang Zhu. "G-Eval: NLG Evaluation using Gpt-4 with Better Human Alignment." In Proceedings of the 2023 Conference on Empirical Methods in Natural Language Processing. Association for Computational Linguistics, 2023.*

---

> > ### Comment · Reviewer_T4KC · 2025-08-04
> >
> > I would like to thank the authors for the detailed rebuttal. I will maintain my score and remain positive towards accepting this work.

---

> > > ### Author Response · Authors · 2025-08-08
> > >
> > > We thank the reviewer for their feedback and for helping us improve the paper, especially the clarity of theoretical results. We will incorporate these changes in the final version and include the comparison between DTS and DTS*.

---

### Official Review · Reviewer_ULya · 2025-07-03

**Clarity:** 3
**Significance:** 3
**Originality:** 2
**Rating:** 4
**Confidence:** 4

**Summary:**

This paper presents Diffusion Tree Sampling (DTS) a inference time method for guidance in diffusion models. DTS adopts MCTS for diffusion models by constructing a tree of possible rollouts using the diffusion chain and backing up rewards for more accurate value estimation of intermediate noisy states. The authors also present a search variant of the method, DTS*, for generating single high quality samples.

**Questions:**

See weaknesses

**Ethical Concerns:**

["NO or VERY MINOR ethics concerns only"]

**Final Justification:**

Based on the response from the authors, I will maintain my rating in favor of acceptance.

**Limitations:**

yes

**Paper Formatting Concerns:**

no concerns

**Quality:**

2

**Strengths And Weaknesses:**

**Strengths**:

1. The method builds on top of MCTS and thus inherits the advantages of MCTS as an anytime algorithm.
2. The method is relatively straightforward to understand.
3. Empirical results on Stable Diffusion are good to have but I have some concerns on other experimental designs (see weaknesses).

**Weaknesses**:

I don't have much criticism around the methodology since the method is pretty clear from the main text and the authors have clarified a lot of expensive design choices in Section 6 (However I think the authors should emphasize the runtime per prompt from App F.3 in more details in this section). I have some followup questions around the experimental design instead which I highlight as follows.

1. How significant are the results on class conditional synthesis like MNIST/CIFAR-10. I feel like these benchmarks are heavily deprecated and do not reveal the complete picture.  Can the authors choose some other tasks. For instance, I would be interested in comparing DTS for inverse problems like super-resolution or in-painting? On these tasks DPS could still be considered a competitive baseline.

2. Second, for the experiments on Stable Diffusion, can the authors explain why Best-of-N is a strong baseline since it feels like the most naive algorithm for any kind of inference time steering for diffusion models? Moreover, based on my understanding, unlike methods which tilt the diffusion guidance towards a pre-specified reward, Best-of-N does not do it. Therefore this observation seems a bit odd.

3. How is exploration controlled in this algorithm? Do you always create a new node when the |C_t| < maximum value or with some probability traverse a chain with a high value estimate?

4. Minor issues: Line 171: Do you mean x_T+1 instead of x_0
5. Lastly there is some existing work [1,2,3] on gradient based steering of diffusion models which relies on similar ideas of tilting the reward distribution while simultaneously learning the guided posterior directly. I guess the paper can benefit from additional discussion around the limitations of these works.

[1] Variational Control for Guidance in Diffusion Models - Pandey et al.

[2] RB-Modulation: Training-Free Personalization of Diffusion Models using Stochastic Optimal Control - Rout et al.

[3] Symbolic Music Generation with Non-Differentiable Rule Guided Diffusion - Huang et al.

---

> ### Author Rebuttal · Authors · 2025-07-31
>
> We thank the reviewer for their thorough review and positive assessment of our work. We address your concerns below, and also draw attention to our response to other reviewers for new experiments - text diffusion with LLM evaluation (reviewer 1bSF), comparison between DTS and DTS* (reviewer T4KC), and comparison of memory and runtime (reviewer t7aZ). Note that this year, NeurIPS does not allow posting any links or documents in the rebuttal.
>
> > I think the authors should emphasize the runtime per prompt from App F.3 in more details in this section
>
> We refer to our response to reviewer t7aZ for a comparison of memory usage and runtime per prompt for text-to-image generation using Stable Diffusion 1.5 with ImageReward. We will include these results in the final version.
>
> > How significant are the results on class conditional synthesis like MNIST/CIFAR-10. I feel like these benchmarks are heavily deprecated and do not reveal the complete picture.
>
> While these benchmarks are saturated for tasks like image classification, they are still used in recent works for diffusion inference-time alignment or fine-tuning [1,2] since it is not trivial to sample from class-conditioned distributions using a base model that samples uniformly from all classes. We believe the results are significant, since the performance gap between DTS and the other methods is substantial, especially as we increase inference-time compute. The qualitative samples in Figure 6 and Appendix H.2 also showcase the specific characteristics of samples obtained using different methods. In particular, existing methods seem to miss certain modes or show characteristics of mode collapse, which shows that this task is not trivial for these methods.
>
> > Can the authors choose some other tasks. For instance, I would be interested in comparing DTS for inverse problems like super-resolution or in-painting? On these tasks DPS could still be considered a competitive baseline.
>
> Following the reviewer’s request, we ran an inpainting experiment on CIFAR10 images with a Gaussian forward channel as in the DPS paper. Due to time limitations, we run this experiment on 20 images (two images sampled randomly from each class) and average the results. For each 32x32  image, we mask the 10x10 patch from the center and use the negative error between the generated sample and the unmasked pixels from the original image as the reward. We report the final average MSE between 16 generated images and the reference image.
>
> The results below show that DPS, which uses the gradient of the reward, is the only method that obtains good results on this task. We hypothesize this is because the inpainting task has a very sharp posterior where the sampled image must exactly match the unmasked pixels (compared to class conditional sampling where the posterior represents a semantic concept and has a wider mode). This is a very hard search problem and without the gradient, there is very little signal in most regions of the pretrained model distribution.
>
> *Avg. MSE between generated images and reference image, lower is better*
> | **Method / NFEs**   | **10000** | **25000** | **50000** | **100000** | **250000** | **500000** |
> |:---------|----------:|----------:|----------:|----------:|----------:|----------:|
> | **DPS**      | **0.01806** | **0.01563** | **0.01502** | **0.01501** | **0.01607** | **0.01773** |
> | **SMC/FK**   | 0.14587  | 0.19196  | 0.14906   | 0.19285  | 0.17591  | 0.19706  |
> | **DTS (Ours)**  | 0.16829 | 0.15937 | 0.15717 | 0.14121 | 0.13013 | 0.13018 |
>
>
> We then incorporate the gradient into DTS, where we use the gradient term in addition to the score model to obtain the next state when constructing the tree. The backup and selection procedure remain the same. With the gradient-guided proposal, DTS improves upon DPS and even SMC + gradient (which is the same as TDS). We also observe it allows the performance to consistently improve with more compute, whereas vanilla DPS does not improve with more compute. At 500k NFEs, DTS + gradient improves **32.96%** compared to vanilla DPS.
>
> *Avg. MSE between generated images and reference image, lower is better*
> | **Method / NFEs**   | **10000** | **25000** | **50000** | **100000** | **250000** | **500000** |
> |:---------|----------:|----------:|----------:|----------:|----------:|----------:|
> | **DPS**      | 0.01806 | 0.01563 | 0.01502 | 0.01501 | 0.01606 | 0.01773 |
> | **SMC+gradient/TDS**     | 0.01415 | 0.01331 | 0.01916 | 0.02086 | 0.01981 | 0.01591 |
> | **DTS+gradient (Ours)**    | **0.01302** | **0.01263** | **0.01227** | **0.01194** | **0.01153** | **0.01189** |
>
>
> > Why is best-of-N a strong baseline?
>
> There are two different types of experiments considered in our paper: (1) posterior sampling experiments (the illustrative 2D and class-conditional image generation experiments), and (2) what one may call “high reward search” experiments (text-to-image and text diffusion experiments).
>
> In the first setting, there is a well-defined and tractable ground truth distribution (the reward-tilted distribution in the 2D case and the class-conditioned image distribution in the image case). Here, we can compare with the ground truth samples to evaluate how well methods like DTS and baselines match the target distribution.
>
> In the second setting, the goal is to produce the single best sample that achieves both high likelihood under the pretrained model and high reward [3,4]. To generate an image for a given caption, for example, we care about having a single good image. In this setting, prior work [3,4] shows that this best-of-N approach is surprisingly strong because the pretrained model already draws from a rich, high-entropy distribution and best-of-N simply cherry-picks the luckiest draw.
>
> Recent work [5] provides theoretical guarantees for best-of-N for test-time scaling of LLMs: under some coverage conditions (if the base model assigns non-negligible probability to every high-reward region), there exists a finite N for which best-of-N achieves optimal performance. This makes best-of-N both a powerful baseline and a useful stress-test for any new inference-time alignment method. Yet the guarantees are brittle - when coverage is imperfect (the realistic case), the bounds loosen and best-of-N may miss entire reward modes; and as N grows large, the best sample increasingly exploits imperfections in the reward model, leading to reward hacking. We believe that our principled search method using unbiased soft values should help with both of these issues, since soft value functions can help guide denoising towards undersampled regions as well as avoid reward hacking.
>
> > How is exploration controlled in this algorithm? Do you always create a new node when the |C_t| < maximum value or with some probability traverse a chain with a high value estimate?
>
> We always expand from a node if the number of children is less than the maximum allowed value. The trade-off between exploration and exploitation is controlled with this maximum value, which depends on the number of visits (this branching technique is called progressive widening in the literature [6]). Since our selection step (whether sampling as in DTS or picking the max as in DTS*) prioritizes higher value nodes, these nodes end up being visited more often and hence are allowed more children during expansion.
> > Minor issues: Line 171: Do you mean x_T+1 instead of x_0
>
> We believe the reviewer is referring to line 177, and it is a typing error - it should be x_{T+1} as suggested. Thanks for pointing that out!
>
> > Lastly there is some existing work [A,B,C] on gradient based steering of diffusion models which relies on similar ideas of tilting the reward distribution while simultaneously learning the guided posterior directly. I guess the paper can benefit from additional discussion around the limitations of these works.
>
> Thank you for suggesting these works. We will add the following discussion in the related works section:
>
> “Gradient-steering works treat guidance as stochastic control: Pandey et al. [A] learn KL-regularised drifts for differentiable rewards but break when high-reward modes lack coverage; Rout et al. [B] propose a training-free method to modify the drift for style transfer in vision; Huang et al. [C] extend to non-differentiable music rules via high-variance REINFORCE. Our search-based alignment does not require differentiability or dense coverage assumptions and is domain-agnostic.”
>
> *[1] Wu et al. "Practical and asymptotically exact conditional sampling in diffusion models." Advances in Neural Information Processing Systems 36 (2023): 31372-31403.*
>
> *[2] Venkatraman et al. "Amortizing intractable inference in diffusion models for vision, language, and control." Advances in neural information processing systems 37 (2024): 76080-76114.*
>
> *[3] Singhal et al. "A General Framework for Inference-time Scaling and Steering of Diffusion Models." In Forty-second International Conference on Machine Learning, 2025.*
>
> *[4] Ma  et al. "Inference-time scaling for diffusion models beyond scaling denoising steps." arXiv preprint arXiv:2501.09732 (2025).*
>
> *[5] Huang et al. "Is Best-of-N the Best of Them? Coverage, Scaling, and Optimality in Inference-Time Alignment." In Forty-second International Conference on Machine Learning, 2025.*
>
> *[6] Couëtoux et al. "Continuous upper confidence trees." In International conference on learning and intelligent optimization, pp. 433-445. Berlin, Heidelberg: Springer Berlin Heidelberg, 2011.*
>
> *[A] Pandey et al. "Variational Control for Guidance in Diffusion Models." In Forty-second International Conference on Machine Learning, 2025.*
>
> *[B] Rout et al. "Rb-modulation: Training-free personalization of diffusion models using stochastic optimal control." arXiv preprint arXiv:2405.17401 (2024).*
>
> *[C] Huang et al. "Symbolic Music Generation with Non-Differentiable Rule Guided Diffusion." In International Conference on Machine Learning, pp. 19772-19797. PMLR, 2024.*

---

> > ### Author Response · Authors · 2025-08-04
> >
> > Dear reviewer, we appreciate your time and feedback on our work. As the discussion period comes to an end, we wanted to confirm if our rebuttal addressed your concerns. We added experiments on the inpainting task and showed that DTS with added gradient guidance outperforms existing methods and improves with more compute. We also answered all of your questions about baselines, exploration, and related works.
> >
> > If there are any outstanding questions, we will be happy to answer them!

---

> > ### Comment · Reviewer_ULya · 2025-08-05
> > **Response**
> >
> > Thanks for your detailed response which addresses most of my concerns. I would recommend the authors to highlight some key findings from their experiments on Inpainting in their response in the limitations section of their work where DTS might not be very suitable. I will maintain my score on this work.

---

> > > ### Author Response · Authors · 2025-08-08
> > >
> > > We thank the reviewer for their feedback and for helping us improve our paper. The inpainting experiments were certainly interesting and helped highlight the relative strengths of different methods.

---

### Official Review · Reviewer_1bSF · 2025-07-05

**Clarity:** 4
**Significance:** 2
**Originality:** 2
**Rating:** 4
**Confidence:** 3

**Summary:**

This paper proposes Diffusion Tree Sampling (DTS) and its greedy variant Diffusion Tree Search (DTS*), for better adapting a pretrained diffusion model to new objectives at inference time. This method adopts a tree-based approach to sample from the reward-aligned target density and refine value estimates with each additional generation. From theory proof to experimental results, DTS and DTS* have shown a promising way for new objectives with less computation and higher performance.

**Questions:**

- Please provide more details on the backup process. And the comparison about the value estimation should be added.
- For text generation, the results do not express the advantages of the proposed method. Please add more evaluation metrics for a full analysis to showcase the superiority of the proposed method.
- Please provide experiments with different numbers of nodes and different numbers of denoising steps.

**Ethical Concerns:**

["NO or VERY MINOR ethics concerns only"]

**Final Justification:**

In the author's response, the clarification about baselines addresses some of my concerns. Results for CoLA reward function are also very helpfu. Therfore, I adjust my rating and lean towards acceptance.

**Limitations:**

Yes

**Quality:**

3

**Strengths And Weaknesses:**

Strengths
- The paper combines RL with the diffusion model and tackles the credit assignment problem during the denoising process. Therefore, DTS can assign not only terminal rewards but also processing rewards.
- The authors provide theoretical guarantees to ensure that DTS produces a sequence of terminal states whose empirical distribution converges to the optimal policy.
- The illustrative experiments, class-conditional posterior sampling, and image and text generation experiments demonstrate the effectiveness of the proposed method.
Weaknesses
- Lack of comparison in value estimation problem: The core idea of the tree-based sampling approach is to use the value of children nodes to update the value of parent nodes, as mentioned in the backup process. However, there are no experiments about how much the differences is between the estimated value from the proposed method and the ground-truth value. And the comparison between other baselines concerning the value estimation.
- Insufficient baselines in terms of image and text generation: The proposed method is only compared with FK/SMC and best-of-N for image and text generation. What about the results of other baselines? The results in Figure 8 between the proposed method and best-of-N in the text generation are almost the same. What is the advantage of the proposed method in this setting?
- The setting of the number of nodes and denoising steps: How to set these two variants, is it an empirical value, or else? Whether the experiment section should include a sensitive analysis.

---

> ### Author Rebuttal · Authors · 2025-07-31
>
> We thank the reviewer for their comments and suggestions. We address your concerns below, and also draw attention to our response to other reviewers for added experiments - image inpainting (reviewer ULya), comparison between DTS and DTS* (reviewer T4KC),  and comparison of memory and runtime (reviewer t7aZ). Note that this year, NeurIPS does not allow posting any links or documents in the rebuttal.
>
> > There are no experiments about how much the differences is between the estimated value from the proposed method and the ground-truth value
>
> Indeed, we believe this is an important point and we do have experiments to support it; based on reviewer’s comment we will highlight this more clearly. We have a bias-variance analysis of the value estimates with respect to the ground truth values in Figure 5. Here, we perform 1000 Monte Carlo rollouts from noisy states using the base model to get a reasonably good approximation of ground truth. Figure 5 shows that the value estimates of DTS have lower bias and variance compared to the baselines, including DPS and SMC + variants like TDS, DAS (which all use Tweedie’s formula) and SMC + rollout (using a single DDIM rollout). We limit this error analysis to the 2D experiments, since estimating the ground truth values is prohibitively expensive in higher dimensions, both due to model size and also because we require exponentially more MC rollouts to reliably estimate the ground truth values. Please let us know if you have an additional kind of evaluation in mind.
>
> > The proposed method is only compared with FK/SMC and best-of-N for image and text generation. What about the results of other baselines?
>
> There are two different settings considered in our paper: (1) posterior sampling (illustrative 2D and class-conditional image generation), and (2) what one may call “high reward search” (text-to-image and text diffusion). The nature of these two distinct tasks and prior literature motivate the choice of algorithms.
>
> In the first setting, there is a well-defined and tractable ground truth distribution. DTS and the baselines in these experiments are designed to sample approximately from a target distribution; hence, we report distribution-based metrics.
>
> In the second setting, the goal is to produce the single best sample that achieves both high likelihood under the pretrained model and high reward [1,2]. To generate an image for a given caption, for example, we care about having a single good image. DTS and the other baselines from the previous setting are not designed to produce a single good sample (and the corresponding papers also consider the sampling setting only). Therefore, we only use baselines from prior works that report results for a single best sample. The reviewer may be interested in our comparison of DTS and DTS* on text-to-image generation in our response to reviewer T4KC.
>
> > The results in Figure 8 between the proposed method and best-of-N in the text generation are almost the same. What is the advantage of the proposed method in this setting?
>
> > For text generation, the results do not express the advantages of the proposed method. Please add more evaluation metrics for a full analysis to showcase the superiority of the proposed method.
>
> For text generation, even small differences in reward values can result in qualitatively distinct text samples. The difficulty of text evaluation has been discussed in previous works [3,4], which show that metrics generally correlate poorly with human judgments and do not necessarily follow a linear scale. Below, we score different methods with ChatGPT 4o, which is again not a perfect metric, but aligns closer with human evaluation [5].
>
> The fact that FK/SMC produced overly repetitive outputs is also primarily attributed to issues with these reward functions. DTS* uses soft values that accumulate posterior mass and can balance likelihood under the pretrained MDLM with optimizing the reward, producing better quality samples due to the influence of MDLM. Following the reviewer’s observation, the difference between DTS* and FK/SMC in the text experiment also seems small in magnitude, but the samples in Figure 8 differ significantly.
>
> We ran an additional experiment using a reward function trained on Corpus of Linguistic Acceptability (CoLA) [6], and we observed a similar phenomenon, see results below. The absolute difference between DTS* and baselines is again small, but the difference in text quality is significant.
>
> **Results for CoLA reward function**
>
> *CoLA rewards, higher is better*
> | **Method / NFEs** | $2^9$ | $2^{11}$ | $2^{13}$ | $2^{15}$ | $2^{17}$ |
> | :--- | ---: | ---: | ---: | ---: | ---: |
> | **DTS\*** | **-0.262202** | **-0.141963** | **-0.074698** | **-0.045357** | **-0.035025** |
> | **Best-of-N** | -0.382968 | -0.180514 | -0.100434 | -0.063361 | -0.049272 |
> | **FK/SMC** | -0.296245 | -0.200835 | -0.120380 | -0.088619 | -0.056350 |
>
>
> **Prompt: “Once upon a time”**
> - **DTS\* (reward = -0.027):** Once upon a time, I texted you and said,”It’s going really well.” I’d say,”I suppose you would be surprised, wouldn’t you?” Now, a few days ago, you were texting me, and
> - **Best-of-N (reward = -0.037):** Once upon a time, Samuel Adams was a famous launch merchant. Now I am afforded opportunity to observe the financial earnings of our Founders, President, and reminds me, at the @Beauty Post. I find myself very excited about & very interested in game mechanics, character models, jurists, movements
> - **FK/SMC (reward = -0.075):** Once upon a time women would commit themselves into fleets, spawning “Stupid”, STupid”, “Stupid”, “ST”, “Stupid”, “Stupid”, “Stupid”, “St
>
>
> **Prompt: “The pizza”**
> - **DTS\* (reward = -0.027):** The pizza was topped with two teriyaki balls. These huge, tomatoey balls looked like they came from a pizza box. The dough was kind of soft and they balanced nicely. The vegetable sauce was voluminous. Another topping feature was ricotta. Ricotta uses a soft crust
> - **Best-of-N (reward = -0.038):** The pizza was made primarily of a cheese seasoned with sugar and spices. Image caption Cardiffers were first to sip their new beer on Boxing Day in 2016. Cardiffers were first to taste its brand new beer when it opened Boxing Day in 2016. Its doors opened when the 3200-seat
> - **FK/SMC (reward = -0.072):** The pizza is described as follows: B.298.50.57.16.99.1202.7.4601.8.069.5.1105.15.1113.16.5916914513.352.28017.28018.36.43
>
>
> **LLM evaluation**
>
> We provide text samples to ChatGPT 4o and asking it to score them between 0 and 1 following a rubric (the prompt is omitted due to the character limit; we can provide it if the reviewer wishes). Results are averaged over 3 completions for each of the 15 prompts for $2^{17}$ NFEs. We observe that DTS\* improves **31.3%** over Best-of-N when using inifini-gram reward, and **13.7%** when using CoLA reward.
>
> *LLM evaluation of text samples, higher is better*
> | **Method / Reward** | **infini-gram** | **CoLA** |
> |:---|---:|---:|
> | **DTS\*** | **0.302 ± 0.023** | **0.322 ± 0.022** |
> |  **Best-of-N** | 0.230 ± 0.022 | 0.283 ± 0.020 |
> | **FK/SMC** | 0.111 ± 0.014 | 0.216 ± 0.019 |
>
>
> > The setting of the number of nodes and denoising steps
>
> We do not explicitly specify the number of nodes for the tree. Instead, we specify a maximum number of diffusion model evaluations (NFEs reported in all of our figures) for a fair comparison across methods. For a concrete example, at $10^6$ NFEs with 50 denoising steps per sample, best-of-N and the SMC-based methods would use 20000 particles, and DTS/DTS* would have $10^6$ total nodes. All of our experiments compare these methods across different NFEs to understand how they scale with more compute (or equivalently, how they compare when varying the number of particles/nodes).
>
> The number of denoising steps is orthogonal to the method used for inference-time alignment, and generally increasing the denoising steps results in higher quality samples from the base model, which is common across all methods. We use typical values of denoising steps for all methods: 50 is the default for pre-trained image and text-to-image models, and for text diffusion, it depends on the sequence length.
>
>
> > Please provide more details on the backup process
>
> The backup process follows Equation 5, where the value of a state $x_t$ is the log-mean-exp of the value at states $x_{t-1}$, where the mean is over samples from the pretrained model $p_\theta$. In DTS/DTS*, this translates to the value of each node being the log-mean-exp of its children values; since the children are sampled using $p_\theta$, the expectation is over $p_\theta$ as in equation 5. During tree construction, we traverse the tree until we reach a terminal node. We set the value of the terminal nodes equal to the exponent of the reward, and then iteratively update the values of the parents along this path. We hope this explanation helps clarify the backup, we will be happy to answer any further questions.
>
>
> *[1] Singhal et al. "A General Framework for Inference-time Scaling and Steering of Diffusion Models." In Forty-second International Conference on Machine Learning, 2025.*
>
> *[2] Ma et al. "Inference-time scaling for diffusion models beyond scaling denoising steps." arXiv preprint arXiv:2501.09732 (2025).*
>
> *[3] Hashimoto et al. "Unifying Human and Statistical Evaluation for Natural Language Generation." In Proceedings of the 2019 Conference of the North American Chapter of the Association for Computational Linguistics, Volume 1 (Long and Short Papers), pp. 1689-1701. 2019.*
> *[4] Celikyilmaz et al. "Evaluation of text generation: A survey." arXiv preprint arXiv:2006.14799 (2020).*
>
> *[5] Liu et al. "G-Eval: NLG Evaluation using Gpt-4 with Better Human Alignment." In Proceedings of the 2023 Conference on Empirical Methods in Natural Language Processing, 2023.*
>
> *[6] Warstadt et al. "Neural network acceptability judgments." Transactions of the Association for Computational Linguistics 7 (2019): 625-641.*

---

> > ### Author Response · Authors · 2025-08-01
> > **Prompt used for LLM evaluation**
> >
> > Prompt used for LLM evaluation of text samples:
> >
> > """
> > You are an expert language evaluator. Your task is to assess the overall quality of a text sample based on the following criteria:
> >
> > 1. **Fluency**: Is the text generally grammatically correct and understandable?
> > 2. **Diversity**: Does the text show some variety in vocabulary or sentence structure, even if modest?
> > 3. **Plausibility**: Is the content mostly coherent, logical, and believable in its context?
> >
> > Please be constructive and fair in your evaluation — highlight the strengths of the sample as well as areas that could be improved. Some minor errors or simplicity are acceptable if the overall message is clear. You should penalize responses that use too many non-linguistic symbols.
> >
> > For each sample, provide a brief explanation touching on each criterion, followed by a quality score between **0.0** (very poor) and **1.0** (excellent). Format the final score using `\\boxed{{score}}`.
> >
> > ---
> >
> > **Example 1 (High Quality)**
> >
> > Sample:
> > \"\"\"The shimmering lake reflected the golden hues of the setting sun as birds glided across the surface, their wings slicing through the calm evening air.\"\"\"
> >
> > Evaluation:
> > - **Fluency**: The sentence flows smoothly with proper grammar and elegant phrasing.
> > - **Diversity**: The vocabulary is rich ("shimmering", "glided", "golden hues"), and sentence structure is varied.
> > - **Plausibility**: The description is vivid and realistic.
> > \\boxed{{0.95}}
> >
> > ---
> >
> > **Example 2 (Low Quality but Understandable)**
> >
> > Sample:
> > \"\"\"Lake is shiny. Birds is fly. The sky is color like orange. It is lake.\"\"\"
> >
> > Evaluation:
> > - **Fluency**: Some grammatical issues ("birds is fly", "it is lake"), but basic meaning is understandable.
> > - **Diversity**: Limited vocabulary and repetition.
> > - **Plausibility**: The message is simple but not nonsensical.
> > \\boxed{{0.25}}
> >
> > ---
> >
> > Now evaluate the following sample:
> >
> > Sample:
> > \"\"\"{sample}\"\"\"
> >
> > Evaluation:
> > """

---

> > > ### Author Response · Authors · 2025-08-04
> > >
> > > Dear reviewer, we appreciate your time and feedback on our work. As the discussion period comes to an end, we wanted to confirm if our rebuttal addressed your concerns. We addressed all of your concerns - added new text diffusion experiments and LLM evaluation as requested, and clarified that the paper does include experiments that compare bias-variance of value estimates with other methods, and all of our experiments include varying the number of nodes/particles.
> > >
> > > If there are any outstanding questions, we will be happy to answer them!

---

> > > > ### Comment · Reviewer_1bSF · 2025-08-05
> > > > **Response to author rebuttal**
> > > >
> > > > Thank you for the detailed response. The clarification about baselines addresses some of my concerns. Results for CoLA reward function are also very helpful. I will adjust my final scoring

---

> > > > > ### Author Response · Authors · 2025-08-08
> > > > >
> > > > > We thank the reviewer for their feedback and for helping improve our paper, especially the text diffusion experiments.

---

### Note · Authors · 2025-08-12

We thank the reviewers and the AC for their thoughtful feedback and time. During the discussion, we added new experiments and clarifications, and all reviews remained positive toward acceptance. Below, we summarize the key updates and outcomes.

### Reviewer 1bSF
We added a text-diffusion experiment with a new reward and evaluated generations using an LLM for human-aligned judgments, as suggested. We also clarified several experimental details in the paper. The reviewer indicated they will revise their score.

### Reviewer ULya
The reviewer was positive about the method and raised thoughtful, experiment-focused questions. We clarified these and added an inpainting study. Methods that do not use gradients (e.g., SMC and vanilla DTS) struggled on this task due to the sharp posterior; with a gradient-aided proposal, DTS improved over other methods and continued to benefit from additional compute. We did not receive further questions during the discussion; the reviewer ultimately maintained their score while remaining positive.

### Reviewer T4KC
We appreciate the detailed, theory-centered review. We confirmed the reviewer’s understanding and provided explicit text that we will add to the paper to improve clarity. Per their suggestion, we compared DTS and DTS*, showing complementary strengths: DTS* is preferable for finding a single high-reward sample while remaining likely under the base model, whereas DTS excels for reward-tilted sampling. We did not receive additional questions during the discussion; the reviewer remained positive towards acceptance and maintained their score.

### Reviewer t7aZ
The main concern was compute. We provided detailed memory/runtime comparisons on text-to-image diffusion showing the trade-offs: DTS uses substantially less memory than SMC, scales better than best-of-N, and matches the performance of the strongest baseline with roughly 2x less wall-clock time. The reviewer indicated they will update their score.

### Closing
Across reviews, our additions strengthened the paper: we expanded experiments (including inpainting), clarified theoretical aspects, and provided concrete efficiency analysis. We believe the contribution is both principled and practically useful, and we are committed to incorporating all suggested improvements. While we could not share images during the rebuttal, we have also improved the presentation of all figures (larger text labels, white background, thicker lines, and larger markers).

---

### Decision · Program_Chairs · 2025-09-17

**Decision:**

Accept (poster)

**Comment:**

This paper proposes an inference-time steering method for diffusion models based on Monte Carlo Tree Search (MCTS). The proposed method is straightforward, principled, and demonstrates strong empirical performance across multiple modalities, including 2D data, image, and text. As acknowledged by the reviewers, the paper is clearly written and well-structured. The discussion primarily addressed clarifications regarding the method, theoretical aspects, and experimental setup. All the reviewers found the response from the authors satisfactory. Overall, this paper is a solid contribution, and thus I recommend acceptance.

One important note is the existence of very recent related work [1], which also applies MCTS to diffusion models. Although there are clear differences in implementation and application scenarios, it would be helpful to mention and discuss these distinctions in the revised manuscript.

[1] Jaesik Yoon, Hyeonseo Cho, Doojin Baek, Yoshua Bengio, Sungjin Ahn. Monte Carlo Tree Diffusion for System 2 Planning. ICML 2025.